# Anti-Glucotoxicity Effect of Phytoconstituents *via* Inhibiting MGO-AGEs Formation and Breaking MGO-AGEs

**DOI:** 10.3390/ijms24087672

**Published:** 2023-04-21

**Authors:** Neera Yadav, Jyoti Dnyaneshwar Palkhede, Sun-Yeou Kim

**Affiliations:** 1College of Pharmacy, Gachon University, #191, Hambakmoero, Yeonsu-gu, Incheon 21936, Republic of Korea; 2School of Medicine, Kyung Hee University, Dongdaemun-gu, Seoul 02447, Republic of Korea; 3Department of Chemistry, College of Pharmacy, Gachon University, #191, Hambakmoero, Yeonsu-gu, Incheon 21936, Republic of Korea

**Keywords:** AGEs, MGO, hyperglycemia, phytoconstituents, wound healing, glucotoxicity

## Abstract

The therapeutic benefits of phytochemicals in the treatment of various illnesses and disorders are well documented. They show significant promise for the discovery and creation of novel medications for treating a variety of human diseases. Numerous phytoconstituents have shown antibiotic, antioxidant, and wound-healing effects in the conventional system. Traditional medicines based on alkaloids, phenolics, tannins, saponins, terpenes, steroids, flavonoids, glycosides, and phytosterols have been in use for a long time and are crucial as alternative treatments. These phytochemical elements are crucial for scavenging free radicals, capturing reactive carbonyl species, changing protein glycation sites, inactivating carbohydrate hydrolases, fighting pathological conditions, and accelerating the healing of wounds. In this review, 221 research papers have been reviewed. This research sought to provide an update on the types and methods of formation of methylglyoxal-advanced glycation end products (MGO-AGEs) and molecular pathways induced by AGEs during the progression of the chronic complications of diabetes and associated diseases as well as to discuss the role of phytoconstituents in MGO scavenging and AGEs breaking. The development and commercialization of functional foods using these natural compounds can provide potential health benefits.

## 1. Introduction

The methylglyoxal pathway is the negative side of the glycolysis pathway. It aids in the formation of the highly active dicarbonyl metabolites methylglyoxal (MGO) and glyoxal (GO), which induce the formation of advanced glycation end products (AGEs). The production of AGEs in food is accelerated when it is grilled, broiled, roasted, seared, or fried [1]. The method of heating and the temperature during food preparation has a great impact on the quality of food that may induce the formation of a variety of α-dicarbonyls. These dicarbonyls can further modify lysine and arginine residues in proteins to generate AGEs [2]. Studies showed that during early drying, the formation of Amadori products was higher due to the high temperature and intermediate moisture level; however, under continual heating, Amadori products underwent subsequent reactions [3]. Scientists gave these the term dAGEs, for dietary advanced glycation end products (AGEs from meals), in the human body to distinguish them from endogenous AGEs. dAGEs are basically a class of thermal processing chemicals that are produced by the heat treatment of food products. The most common AGE in food is N(6)-carboxymethyllysine (CML) [4]. MGO is quickly accumulated and aggregated in many tissues. The cellular concentration of the most active and plentiful glycating agent, MGO, ranges from 1 to 5 μM and, in extreme cases, measurements of >300 μM have been identified [5]. MGO can promote endothelial dysfunction (ED); microvascular complications such as nephropathy, retinopathy, and neuropathy; and macrovascular complications such as atherosclerosis, heart failure, impaired revascularization, and impaired wound healing. Additionally, MGO can promote obesity, insulin resistance, and age-related diseases such as hypertension, cancer, and neurodegeneration including dementia, Parkinson’s disease (PD), schizophrenia, and anxiety disorders. The presence of MGO causes dysfunction in three microvascular tissues, including the peripheral nervous system, the eyes, and the kidneys, which are thought to be typical target tissues in diabetes. MGO-affected endothelial functions can change the gene expression linked to coronary artery disease, the upregulation of collagen expression, the activation of apoptotic and inflammatory processes, as well as the reduction of glyoxalase 1 (GLO I) activity in endothelial cells [6]. Research by Zadeh and Yaylayan demonstrated the ability of histidine derivatives to scavenge thermally produced 1,2 dicarbonyl compounds from model Maillard reaction systems. The research highlighted the significance of in situ production of carbonyl scavenging agents as a promising system to reduce the accumulation of toxic compounds in food products [7]. Evidence is being revealed that advocates for encouraging the antidiabetic response of certain newly discovered plant-derived bioactive medicines. Traditional medicine has a promising future in the treatment of diabetes mellitus and has a solid track record in clinical practice. 

The degree of hyperglycemia, the level of oxidative stress in the cell, and the rate at which proteins are turned over for glycoxidation are all substantial contributors to the development of AGEs. Extracellular and intracellular proteins may be both glycated and oxidized in the presence of one or more of these factors. A small fraction of the absorbed simple sugars such as glucose, fructose, and galactose can undergo glycations in the bloodstream, and the process of glycation occurs via the Amadori reaction, the Schiff base reaction, and the Maillard reaction (MR), which result in AGE formation. However, the process of glycation is typically random. Glycation can change the primary, secondary, and tertiary structures of the proteins that can create lysine–arginine crosslinks, which cause protein aggregation. Protein aggregation can affect the feasibility of lysine or arginine groups in the protein that can change the overall glycation process. Additionally, the binding of certain groups at the charged side chain of lysine can also affect the rate of glycation. For example, the attachment of carbohydrate to the lysine side chain by glycation might have an even larger effect because not only the charge but also the size of the side chain of lysine is modified. Additionally, MGO can undergo the nonenzymatic glycation Maillard reaction with proteins, usually at lysine and arginine residues, to form AGEs. In the early stages of MR, condensation reactions between the electrophilic carbonyl group of a reducing sugar and free amino groups, primarily lysine or arginine, result in nonstable compounds known as Schiff bases [8]. The carbonyl group of the sugar molecule combines with an amino group of an amino acid in the first stage of the MR. The end outcome is an imine or a Schiff base, which either separates into its educts or undergoes the Amadori rearrangement to become a stable ketoamine or glycosamine, known as an Amadori product. This equilibrium reaction is more likely to switch to the Amadori product when hyperglycemia becomes more severe and lasts longer. A Schiff base and Amadori product can be formed in any direction, but an actual AGE cannot be created in either direction. These dicarbonyls are eventually converted into AGEs in the final stage of the Maillard reaction as a result of other chemical processes, particularly oxidation and interactions between precursors and reactive oxygen species (ROS). Exogenous AGE intake from tobacco products or food in the form of free adducts is also a possibility, as is the formation of AGEs by oxidative and carbonyl stress [9,10]. The second route of MGO formation is the autoxidation of hexoses, where MGO is synthesized via isomerisation with consequent fragmentation of sugars by retro-aldol condensation. Foods with high hexose content, such as honey, undergo this process [11]. The third route is the oxidation of polyunsaturated fatty acids (PUFA) present in lipids [12]. This pathway produces dihydroperoxide (DHP), which is then cleaved into 6-hydroperoxy-2,4-heptadienal and 4-oxo-2-pentenal. Foods abundant in PUFA, including fish oils, have a high amount of MGO because the 4-oxo-2-pentenal serves as a direct precursor to its production [13]. Zhang et al. has quantified nine AGEs, including argpyrimidine, three isomers of MG-Hs (MG-H1/3), three isomers of GO-hydroimidazolones (GO-Hs), CEL, and CML in black tea. They found arginine derived-AGEs as the major components, especially MGO-derived hydroimidazolones MG-H1/3 (39.66 ± 2.61) μg/g and (58.88 ± 1.76) μg/g, after 1 h drying process of tea leaves [14]. MG-H1 regulates hyperfeeding through tyramine signaling regulated by GATA transcription factor ELT-3 and increases food intake in *Caenorhabditis. elegance* [15]. MGO serves as a precursor during food processing to produce a variety of flavors, such as pyrazines, furanones, pyrroles, thiazoles, pyridines, thiazolines, thiophenes, oxazolines, and oxaoles [16,17,18]. The flavor of many foods, including cheese, wine, and beer, depends on the capacity of MGO to produce aroma molecules in mild conditions [19].

## 2. AGEs Derived from MGO

In the cell, reducing carbohydrates, primarily glucose and carbonyl molecules, react nonenzymatically with proteins, DNA, and lipids. Proteins or lipids that have been exposed to sugars and subsequently become glycated are known as AGEs. AGEs are persistent byproducts of the glycolysis process that play a prominent role in aging and pathogenesis of a number of degenerative diseases, including Alzheimer’s disease (AD), diabetes, atherosclerosis, chronic renal disease, and atherosclerosis. So far, more than 25 different AGEs have been identified in human blood and tissues and in foods. AGEs have been classified into many groups according to their chemical structure and fluorescence-emitting capacity [20]. These can be identified as below:Nonfluorescent crosslinked;Fluorescent and crosslinked;Fluorescent non-crosslinked;Nonfluorescent and non-crosslinked.

Pentosidine, pentodilysine, crossline, AGE-XI, vesperlysine A, and vesperlysine C are examples of naturally fluorescent crosslinked AGEs. The most common AGE structures that are nonfluorescent crosslinkers are imidazolium dilysine crosslinks, often referred to as glyoxal-lysine dimer (GOLD) or methylglyoxal-lysine dimer (MOLD) crosslinks [21]. These structures are produced by the reaction between two lysine sidechains and two molecules of GO or MGO, respectively. Crosslinks of arginine, known as the imidazolium crosslink derived from methylglyoxal and lysinelysine (MODIC) and the imidazolium crosslink derived from glyoxal and lysine-arginine (GODIC), have been isolated from bovine serum albumin (BSA) [22]. CML, carboxyethyl-lysine (CEL), pyrraline, and imidazolones are among the important nonfluorescent and non-crosslinked AGEs [23]. In addition to crosslinked AGEs, diabetic patients’ blood also contains a variety of fluorescent non-crosslinked AGEs. They share a similar structural design to fluorescent crosslinked AGEs, with the exception of the N-H bond in place of one of the bonds that connects the heterocyclic portion to the amino acid. Despite having different chemical structures, all AGEs have lysine residue in their molecules as a common feature (Table 1).

A significant amount of the ingested AGEs remains present in the body and attache inadvertently to proteins, particularly in the liver and kidney tissues. The renal excretion of AGEs is believed to be about 30% of the absorbed amount in healthy people, and this ratio drops to 5% in kidney patients [20]. As a result, dAGE control has a significant impact on human health, including improvements in wound healing, tissue regeneration, insulin resistance, and cardiovascular diseases [44].

AGEs play an important role in diabetes and the related complications, such as retinopathy, nephropathy, cardiovascular diseases (CVDs), and stroke [45,46]. Furthermore, excessive AGE formation permanently damages the structural and functional integrity of macromolecules. These damages trigger a number of age-related complications such neurodegenerative disorders, reduced tissue regeneration, diabetes, and associated complications. Certain carbonyls can also be created by the fatty acid metabolism through lipid peroxidation, which interacts with proteins, DNA, and lipids to form advanced lipoxidation end products (ALEs). CML; pentosidine; hydroimidazolone produced from MGO; malondialdehyde (MDA); and hydroimidazolones derived from GO, such as Nγ-(5-hydro-4-imidazolon-2-yl)ornithine (GO-H1), 5-(2-amino-5-hydro-4-imidazolon-1-yl)norvaline (GO-H2), and 5-(2-amino-4-hydro-5-imida-zolon-1-yl)norvaline (GO-H3), and MGO, such as G-H1, Nε-(5-hydro-5-methyl-4-imidazolon-2-yl)-L-ornithine (MG-H1), 2-amino-5-(2-amino-5-hydro-5-methyl-4-imidazolon-1-yl) pentanoic acid (MG-H2), 2-amino-5-(2-amino-4-hydro-4-methyl-5-imidazolon-1-yl) pentanoic acid (MG-H3), and 3DG-H, are well-known AGEs and AGE precursors. Other AGEs include 3-deoxyglucosone (3DG) derivatives, including 3-deoxyglucosone-derived imidazolium crosslink (DOGDIC) and pyrraline. As a result of the metabolism of high-carbohydrate diets, glycation frequently comprises the alteration of the guanidine group of arginine residues with GO, MGO, and 3-DG. Triose phosphate intermediates in aerobic glycolysis spontaneously decompose in vivo to produce MGO. Usually, glucose has the slowest rate of glycation in contrast to intracellular carbohydrates such as fructose, glucose 6-phosphate, and threose, which have a higher rate of oxidation [47]. It could potentially develop as a result of the oxidative breakdown of lipids and both pentoses and ascorbate, two types of carbohydrates (arachidonate). MGO combines with protein lysine residues to create N(epsilon)-(carboxyethyl)lysine (CEL) and the imidazolium crosslink, methylglyoxal-lysine dimer (MOLD), in addition to reacting with arginine residues to create imidazolone adducts. Moreover, two argypyrimidne AGEs, tetrahydropyrimidine and 5-methylimidazolone, have been identified as MG-arginine adducts [38]. According to research, adding divalent iron to a glucose/alanine model system (a Maillard model system) can result in the synthesis of bis[N,N’-di-glycated alanine]iron(II) complexes and the release of more reactive N,N-diglycated alanine derivatives (mono-glycated Amadori compounds) into the reaction mixture [48]. HbA1c (glycated hemoglobin) is the most common product, formed through the joining of a valine residue present in one of the β chains of the hemeprotein of hemoglobin (Hb) with plasma glucose. The coloring of AGEs is brownish-yellowish, and some of them, such as pyrrolidine, CML, imidazoline, and pentosidine, have fluorescent properties. Pyrraline is produced through a reaction between protein lysine residues and glucose. There are several ways that CML can develop, including the condensation of glucose with lysine from the amino group and sequential rearrangements of the Amadori product, which results in CML through oxidation. Another mechanism involves a direct interaction between GO and the lysine of theamino group [49]. The glycation process in normal conditions takes weeks to years to complete under physiological conditions, but in some pathological conditions, including hyperglycemia, oxidative stress, and temperature rises, the required time might be as little as a few hours.

## 3. How Does MGO Formation Affect the Human Body?

The formation of AGEs and their precursors occurs less frequently during regular metabolic activities of a healthy body, whereas it happens more frequently in chronic diseases such as diabetes and atherosclerosis. In addition, many environmental variables, such as the Western diet, cigarette smoke, and inflammation, enhance AGEs formation. For instance, smoking causes the buildup of additional AGEs in smokers’ plasma due to the high amounts of GO and MG found in cigarette smoke [49]. When AGEs are produced in excess, there is an imbalance between their endogenous production and external intake, efficiency of the AGEs detoxification system, and excretion from the body [50]. Numerous studies have shown that AGEs accumulation leads to oxidative stress, inflammation, and a cumulative metabolic burden that includes both hyperglycemia and hyperlipidemia [51,52]. With an accelerated rate, AGEs production and accumulation are seen to proceed with both diabetes and normal aging. It is established that the relationship between AGEs and the receptor for AGEs (RAGEs) has a significant impact on how diabetic vascular disease develops. Problems caused by the development of oxidative stress in different cell types are linked to vascular inflammation stimulation, thrombosis, and the activation of platelets. AGEs may distract platelet aggregation and fibrin stabilization, resulting in a tendency to thrombogenesis that favors retinopathy in diabetics. Additionally, the accumulation of AGEs in thicker, larger, and enlarged glomerular basement membranes, severe disease nodular lesions, and the mesangial matrix has remarkable effects on diabetic nephropathy. Moreover, it is thought that MGO- and MGO-originated AGEs change the structure and functionality of a number of tissues and organs in the human body. Recent research has shown that the gut microbiota regulates an aging-related loss in gut barrier integrity. Aging usually brings rapid changes and deterioration in the gut microbiome that results in an increased intestinal permeability and thereby an increased absorption of AGEs into the bloodstream that allows for the accumulation of AGEs in microglial tissue, resulting in declined microglial function. There is a potential correlation between age and the amount of CML found in human blood samples [43]. Redox active transition metals, ROS, and oxygen enhance AGE formation.

Advanced glycoxidation end products are usually produced when an oxidative process is involved. Chemically, AGEs are extremely diverse in nature. Apart from the fluorescent property, AGEs possess different crosslinking properties based on their chemical structure. Nonfluorescent non-crosslinking AGEs are formed much faster than other AGEs; therefore, their concentration is usually higher. Moreover, the fluorescent properties of AGEs can be utilized to measure AGEs (biomarker) as skin autofluorescence (SAF) by the AGEs Reader. Numerous AGEs have been found since the first glycated protein, i.e., HbA1c, was discovered. Some of them exhibit distinctive autofluorescence traits, making it easier to identify them in situ or in vivo. Ahmed et al. (2003) studied the concentrations of MG-H1 and -H2 in soluble human lens proteins and compared them to pentosidine and other AGEs concentrations. They detected MG-H as a significant AGEs in human lens proteins. It was observed that MG-H1 concentrations increased by 85%, MG-H2 by 122%, argpyrimidine by 255%, and pentosidine by 183% in cataractous lenses compared to noncataractous lenses [53]. MGO alteration of lens crystallins resulted in reduced arginine residues and thus reduced the positive charge in lens proteins [54]. Up to 2% of the total arginine may be changed by the production of the adducts MG-H1 and -H2. The isoforms of human lens crystallin include 10 to 20 arginine residues. As a result, it is estimated that 10–20% of crystallin molecules can undergo MG-H alteration. The R58H mutation in γ-d-crystallin showed that the loss of a single arginine residue may promote cataract formation [55]. Although transient, MG-H-mediated changes in lens protein charge may cause protein refolding and promote long-lasting irreversible alterations such as oxidation and proteolysis that are linked to cataractogenesis [53]. MGO has a significant role in the formation of representative volatiles in baked and fried foods, despite its harmful effect on the development of food toxins. The coexistence of beneficial and harmful effects of MGO, as well as its involvement in a variety of reactions, makes it difficult to manage MGO and its derivative toxins that directly or indirectly affect human health. Several in vitro and in vivo studies have demonstrated the interaction of various macromolecules with MGO and MGO-mediated molecular mechanisms that contribute to delayed tissue regeneration and help disease progression. These molecular pathways can provide potential therapeutic targets for natural compounds of clinical importance.

### 3.1. Animal Studies

The unique amino acid sequence as well as the folding of the protein structure at secondary, tertiary, and quaternity levels retain all the mechanisms in coordination to maintain normal cellular functioning and homeostasis. However, glycation may change the post-translational modifications of the proteins that may result in unbalanced homeostasis and ultimately disease onset. The amino acid sequence of a protein is the main factor that determines whether a site will become glycated; however, research has shown that both sequence and folded structure shape the specific complement of AGEs formed. AGEs can cause vascular stiffening by inducing the crosslinking of collagen. Likewise, AGEs can induce the glycation of low-density lipoprotein (LDL), which encourages its oxidation. Oxidized LDL is one of the primary contributors to atherosclerosis because oxidized LDL can be entrapped in the artery walls easily. Additionally, AGEs can bind to RAGEs and trigger oxidative stress in vascular endothelial cells and activate inflammatory pathways [56]. Vascular inflammation is a major contributor to atherosclerosis in mouse models, and it is exacerbated in diabetic macrovessels due to the presence of higher levels of RAGE ligands. RAGE is an interesting target for therapeutic intervention in unusual inflammatory mechanisms and atherosclerosis. Antagonism and genetic disruption of RAGE in atherosclerosis-susceptible mice dramatically reduce vascular inflammation and atherosclerotic lesion area and complexity, suggesting a possible role of RAGEs in atherosclerosis pathogenesis. Ingestion of large amounts of AGEs obtained from diet is a substantial cause of DN in T1D and T2D mice. Avoiding dAGEs offers mice long-lasting protection from DN, which gives the justification for comparable investigations in diabetic patients [57].

In a recent study, MGO altered endothelial function through the receptor-mediated activation of vascular cells by the MGO-arginine adduct hydroimidazolone, and intracellular modifications of protein by MGO in aged diabetic patients have been demonstrated [58]. However, the exact mechanism of MGO-mediated progression of vascular complications in diabetic patients still remains to be uncovered. Increased serum levels of MGO-derived hydroimidazolone AGEs have been identified in patients with type 2 diabetes [59]. The relationship between glucose levels and glycated Hb in diabetic patients can be determined by the hemoglobin glycation index (HGI) [60]. Enhanced ROS production was measured in an STZ-induced diabetic rat model when hemoglobin was altered by MGO in rats [61]. Accumulation of MGO can promote inflammation and vascular damage in diabetic conditions. Diabetes-associated cardiovascular diseases are usually connected to the dysfunction of endothelial cells mediated by MGO in C57BL/6 mice. A number of endogenous pre-AGEs and AGEs-detoxifying systems exist, the most important of which is the GLO enzyme system, which changes the highly reactive glycating dicarbonyl MGO into the less hazardous D-lactate. The genetic capacity (changes in gene expression of transcription factors related to AGEs detoxify mechanisms) to detoxify mechanisms against the accumulation of AGEs may also have an impact on variations in circulating AGEs. Therefore, the amount of AGEs in an organism depends on both the pace at which they are formed and their capacity to be eliminated by natural detoxification mechanisms. Reduced glutathione (GSH), which catalyzes the conversion of GO and MGO to the less toxic D-lactate, is one of several potential detoxifying pathways against AGEs [62]. Another important enzymatic system includes fructosamine kinases that can cause Amadori products to spontaneously break down by phosphorylating and destabilizing them [63]. It was observed that hyperglycemia amplified the circulating inflammatory markers in wild-type but not in MGO-metabolizing enzyme GLO1-overexpressing mice. Additionally, a reduced number of endothelial cells was observed in WT-diabetic hearts than the nondiabetic controls, and GLO1 overexpression preserved capillary density. In the hearts of GLO1-diabetic mice, less myocardial cell death was observed, possibly due to neuregulin production, dimerization of endothelial NO synthase, and Bcl-2 expression in endothelial cells. In GLO1-diabetic mice, lower levels of receptor for AGEs and TNF-α were also evident [64].

### 3.2. Wound Healing and Tissue Regeneration

A growing body of research suggests that the development of AGEs may be a significant factor in the poor wound healing seen in diabetic individuals [65]. Q. Wang et al. investigated the involvement of RAGE-expressing macrophages in the failure of wound healing in diabetic mice in their study titled “Blocking AGE-RAGE Signaling Improved Functional Disorders of Macrophages in Diabetic Wound”. By administering anti-RAGE antibodies topically to the wound, the authors attempted to stop AGE-RAGE signaling. They observed a faster rate of wound healing in the group of mice that had been treated with anti-RAGE antibodies. Additionally, immunohistochemistry labeling showed that macrophages’ phagocytic activity had improved [66].

The identification of the potential chemical metabolites and bioactive compounds may be efficiently utilized for anti-glucotoxicity effects. The extracts of *Mimosa pudica* possess antidiabetic and antihyperlipidemic qualities because they significantly reduced the levels of glucose, triglycerides, LDL, VLDL, and total cholesterol in STZ-induced diabetic rats [67]. In spontaneously hypertensive rats, MGO accumulated in the aorta and artery walls due to vascular contractile failure brought on by aging. The study of the role of MGO in ECs from arteries seems more important because patients with diabetes are at a higher risk of developing arterial conditions such carotid artery stenosis, peripheral vascular disease, and coronary heart disease [68].

Diabetes often results in delayed tissue regeneration and wound healing due to complex molecular mechanisms underlying the angiogenesis and diabetes in connection to each other. Low-AGEs diets may be useful for maintaining the essential balance against autoreactive T-cell responses as well as for preventing direct β-cell injury in non-obese diabetic (NOD) mice. A high AGEs intake may induce excess antigenic stimulus for T-cell-mediated diabetes or direct β-cell injury [69]. Through the induction of oxidative stress, AGEs trigger the activation of a number of stress-induced transcription factors, which then results in the generation of inflammatory mediators such cytokines and acute-phase proteins [70].

### 3.3. In Vitro Studies

Multiple researchers have elucidated AGEs-mediated cellular alterations leading to oxidative stress. MGO promotes numerous protein glycations, which in turn stimulate the emergence of cellular abnormalities that result in mitochondrial dysfunction, apoptosis, and cell death, which are linked to ED acceleration. In hyperglycemic conditions, Hb can be modified by MGO, resulting in the formation of Hb-AGEs. The biochemical modification of Hb by dicarbonyl products results in an increased Hb glycation due to the presence of excess glucose in the cells. Researchers have studied the effects of Hb-AGEs in human umbilical vein endothelial cells (HUVECs). In this study, researchers used an AGEs formation assay to check and compare the affinity of dicarbonyls, such as DL-glyceraldehyde (GA), GO, glycoaldehyde dimer (GC), and MGO, for different proteins, including bovine serum albumin (BSA), Hb, human serum albumin (HSA), collagen type-I, and fibrinogen. These proteins were incubated with dicarbonyls for one to nine weeks to form protein-glycation products, synthesis of MGO-Hb-AGEs being the highest among all. It was observed that MGO was capable of inducing some specific structural and conformational changes in Hb. Hb-AGEs at a concentration of 0.5 mg/mL induced significant cytotoxicity, intracellular ROS generation, reduced mitochondrial membrane potential and downregulation of phosphorylated forms of p-38 and JNK genes expression, cleavage of the nuclear enzyme PARP, and increased Bax/Bcl-2 ratio in HUVECs. Hb-AGEs also induced ED by inhibiting the migration and proliferation of HUVECs [71]. MGO mediates this process by releasing iron from Hb that speeds up the production of ROS. Iron is a redox-sensitive metal that, when used improperly in cells, can change the physiological structure of proteins, leading to cell damage. ED is one of the major factors contributing to unbalanced angiogenesis, which can accelerate diabetic complications by altering cell proliferation and migration at a molecular level in addition to multi-organ damage, ultimately leading to the death of the diabetic patient. A study by Liu et al. in 2012 showed that to minimize vein endothelial angiogenesis, MGO increases autophagy and VEGF receptor 2 (VEGFR2) [72]. MGO-induced vascular apoptosis in diabetes complications involve mitochondrial membrane potential impairment, ROS generation, ED, and glucotoxicity.

A traditional medicine containing *Lespedeza Bicolor* species extract was analyzed through HPLC-Q-TOF-MS/MS and HPLC, and 17 chemical components were studied for their MGO inhibitory effect. Among these, some phytochemicals, such as genistein, naringenin, and quercetin, markedly decreased the production of AGEs and enhanced the activity of substances that break down AGEs, which lowered glucotoxicity. These compounds decreased intracellular ROS, demonstrated anti-apoptotic properties, inhibited MAPK signaling, prevented the MGO-induced AGEs, and degraded preformed AGEs with no known cytotoxicity in HUVECs [73]. The anti-glycation property of these compounds may be correlated to their antioxidant properties and/or their ability to trap MGO. Molecular docking analysis of *Masclura tricuspidata* leaf extract has revealed the presence of several isoflavonoids, out of which 16 phytoconstituents, named cudracus isoflavones A-P (1–16), have been found to ameliorate glycotoxicity-induced metabolic diseases by inhibiting α-glucosidase activity and MGO- and GO-induced AGEs formations [74]. In vitro studies by Branka Vulesevic in 2016 demonstrated that MGO and TNF-α promote endothelial cell death, which was accompanied by elevated angiopoietin 2 expression and decreased Bcl-2 expression in human cardiac ECs (HCECs) [64]. Phytochemical investigations of *Gymnema sylvestre* identified three new pregnane glycosides, gymnepregosides R-T (1–3), that were found to be effective in breaking the MGO-AGEs-protein crosslinks [75]. Flavonoids, phenolics, alkaloids, quinone, glycoproteins, saponins, tannins, and coumarins from *Mimosa pudica* have shown potent ability against the hyperglycemic state. An in vitro experiment demonstrated that *M. pudica* ethanol extract has antihyperglycemic effects by inhibiting diabetes-related enzymes such as glucosidase and amylase when compared to acarbose [76]. LCMS/MS analysis of ethanolic extracts of *M. pudica* has identified several compounds, including 3-dioxolane, 2-tert-butyl-2-phenyl-1, 4-phenylbutan-2-ol, 4-(2-phenylethyl)-phenol, ferulic acid, tyrosinamide, 1-naphthalenecarboxylic acid, myoinositol, caffeic acid, p-hydroxybenzoic acid, 2-hydroxy-benzene ethanol, luteolin, apigenin, fisetin, gallic acid, quercetin, jasmonic acid, naringenin, 3-fluoro-p-anisidine, and monoamidomalonic acid. These phytoconstituents have been shown to reduce the glucotoxicity effect by inhibiting AGE formation and breaking MGO- and GO-AGEs in HUVECs [77]. These findings imply that MGO enhances inflammation in diabetes, which culminates in the loss of endothelial cells. This suggests MGO-induced endothelial inflammation as a target for the therapy of diabetic cardiomyopathy [64].

Ferulic acid, a cinnamic acid derivative, is well-known for its ability to reduce inflammation and function as an AGEs inhibitor. Ferulic acid lowers AGEs and is connected to fructosamine, CML levels, protein carbonyl content, and amyloid cross-structure. In order to prevent diseases caused by AGEs in diabetes complications, ferulic acid is regarded as a potent agent against oxidative stress and protein glycation. It has been discovered that apigenin can create AGEs by directly encasing MGO and then producing apigenin-MGO adducts. To stop inflammation and oxidative stress by AGEs in HUVECs, apigenin frequently reduces the production of ROS and inhibits the expression of adhesion molecules and proinflammatory cytokines. The defense mechanism of apigenin suppresses the extracellular signal-regulated kinase 1/2 (ERK)/transcription factor kappa-light-chain-enhancer of activated B cells signal transduction pathway. This pathway triggered by the AGEs–RAGEs interaction and induction of the ERK/transcription factor (erythroid-derived 2)-like 2 pathways upregulates antioxidant defense molecules [78]. Human endothelium-derived cells treated with catechin under conditions of high glucose showed MGO entrapment. It has been established that using naringenin and caffeic acid in combination with other ingredients dramatically prevented AGEs formation [79]. Quercetin inhibits the production of AGE by trapping regions six and eight of the polyphenol A-ring of MGO and GO simultaneously. Quercetin showed inhibitory an effect on BSA-MGO formation, whereas no effect on BSA-GO, suggesting MGO as the primary precursor for albumin glycation and not the GO [80].

### 3.4. Oxidative Stress and Cell Death

The production of highly reactive dicarbonyl products is enhanced by a high intracellular glucose concentration. AGEs bind to their respective receptors RAGEs and produce ROS. Cellular movement and the increase of proinflammatory and prothrombotic molecules are mediated by AGEs-RAGE interaction on cells including monocytes, macrophages, and endothelial cells. Although AGEs are particularly produced in hyperglycemia, the formation of AGEs in environments with oxidative stress and inflammation suggests that these species may, in part through RAGE, contribute to the pathogenesis of atherosclerosis. The development of atherosclerotic plaques, cardiac and arterial stiffness, and ED may be induced by AGEs accumulation and disruption of the NO signal pathway and oxidative stress [81]. Phenolic phytoconstituents isolated from natural sources have been found to be effective in preventing the generation as well as break down of preformed AGEs. Additionally, pretreatment with peanut extracts dramatically reduced the formation of ROS and cell death caused by MGO in HUVECs. By upregulating Bcl-2 expression and downregulating Bax expression, phytoconstituents of peanut extracts inhibited MGO-induced apoptosis as well as MGO-mediated activation of mitogen-activated protein kinases (MAPKs). In summary, the nutrients in peanuts may reduce oxidative stress and guard against ED and consequences of diabetes [82].

Cell development, differentiation, and repair depend on autophagy and apoptosis. Additionally, cellular homeostasis requires the maintenance of autophagy and apoptosis. There are mainly three types of programmed cell death based on morphological data: type I apoptotic cell death (apoptosis), type II autophagic cell death (autophagy), and type III necrotic cell death (necrosis). It has been demonstrated that MGO acts as a barrier against damage by activating autophagy in human brain microvascular EC [83]. A study by Jae Hyuk Lee in 2020 demonstrated the regulatory effects of MGO-induced autophagy and apoptosis on angiogenesis in human aortic EC (HAoEC). MGO increased the number of autophagic vacuoles, flux, and autophagosomes in the HAoEC in a dose-dependent manner. Additionally, MGO inhibited the pro-angiogenic effect, decreased proliferation, migration, and the formation of tube-like structures. MGO blocked the ROS-mediated Akt/mTOR signaling pathway and caused autophagic cell death. By increasing the ratio of cleaved caspase-3 to Bax/Bcl-2 and by activating the ROS-mediated MAPKs (p-JNK, p-p38, and p-ERK) signaling pathway, MGO also induced apoptosis (Lee et al., 2020). Microtubule-associated protein 1A/1B-light chain 3 (LC3) lipidation is needed for autophagy induction. In MGO-treated cells, there is an increase in LC3-II along with an increase in autophagic flux and autophagic vacuoles. MGO treatment has resulted in a decreased expression of p-Akt, p-mTOR, and VEGF-C, suggesting that MGO-induced autophagy inhibits angiogenesis [84].

## 4. Why Natural Products Are Important in MGO Scavenging/MGO-AGEs Breaking?

Since ancient times, people have used different parts of plants or their extracts to prevent or treat a variety of diseases. Traditional medicine derived from plant extracts has often proven to be more accessible, therapeutically efficacious, and associated with comparatively fewer side effects than modern medications. Today, scientists link these benefits to the presence of a variety of phytochemicals in plants. According to the literature, the pharmaceutical industry has recently given much more attention to the use of phytochemical components of medicinal plants. Phytochemicals are physiologically active substances or nutrients that are derived from plants and can be found in food including beans, whole grains, vegetables, and fruits. They are also occasionally referred to as phytonutrients or phytoconstituents. Secondary metabolites derived from plants, such as phytosterols, phenolics, alkaloids, flavonoids, carbohydrates, tannins, saponins, and terpenoids, have a variety of biological properties that are advantageous to people, including their antiallergic, anticancer, antimicrobial, anti-inflammatory, antidiabetic, and antioxidant activities.

Searching for naturally occurring, plant-based, powerful antiglycation medicines is becoming more popular due to safety concerns. The hypothesized mechanisms of edible plants might be linked to their free radical scavenging capacity during the protein glycation process, which is a vicious cycle between oxidative stress and AGEs [85]. The common phenolic compounds in all Brassica vegetable extracts were sinapic acid and p-hydroxybenzoic acid, which inhibited fuorescent AGE formation by up to 67% [86]. For instance, the aqueous leaf extract from *Brassica. rapa* and *B. oleracea* var. gongylodes reduced blood sugar and cholesterol levels as well as enhanced antioxidant status in diabetic rats [87]. Dichloromethane and ethyl acetate fraction of *Camellia nitidissima* Chi (CNC) extract has been found to inhibit AGE formation by 88.1% and 87.5% at 2.5 mg/mL. Interestingly, >96.0% of MGO was scavenged within 12 h by different fractions of CNC [88]. Epicatechins in green tea and theaflavins in black tea have been shown to lower the MGO in model phosphate-buffered solutions that mimic the physiological environment [89]. Phlorotannins from *Fucus vesiculosus* were found to have antiglycation properties due to their capacity to scavenge reactive carbonyls [90]. Procyanidins isolated from berries have been found to contain the most potent active ingredients for scavenging GO and MGO and preventing the formation of AGEs [91]. The polyphenols and guanidines in *Galega officinalis* extracts have shown antioxidant and MGO trapping activity, which can prevent the vascular consequences of diabetes [92]. Effective protein glycation inhibitors and carbonyl scavengers were obtained from berry phytochemicals. BSA-fructose, BSA-MGO, and arginine-MGO models were each strongly inhibited by (+)-catechin and procyanidins isolated from berry extracts. These phytochemicals might help fight against AGEs-related chronic illnesses [91]. Zhang et al. observed that time and temperature were two vital factors responsible for the synthesis of AGEs from lysine and arginine in black tea during the drying process. Therefore, it is possible to develop strategies such as controlling drying temperatures and times to prevent AGEs formation during the drying of tea leaves [14]. However, green tea contains catechins that may prevent the formation of CML and CEL, although the drying process of the tea may lower the effectiveness of these catechins [2].

## 5. Phytoconstituents in Disease Prevention and Treatment

Over 8000 phytochemicals have currently been identified. They fall under the broad categories of carotenoids and polyphenols, but there are numerous further subdivisions. Based on the active ingredients, a wide variety of phytoconstituents have been discovered and they have been divided into 16 main groups. The most significant phytoconstituents are alkaloids, terpenoids, phenols, polyphenols and phenolic glycosides, coumarins and their glycosides, anthraquinones and their glycosides, flavones and flavonoid glycosides or heterosides, mucilage and gums, tannins, volatile oils, saponins, cardioactive glycosides, cyanogenic glycosides, etc. In addition to these, vital nutrients include amino acids, minerals, vitamins, antibiotics, fiber, carbs, organic acids, lipids, and some sugars. Phytoconstituents that have been shown to have antiglucotoxicity and antiglycation activity have been classified as flavonoids, quinones, phenols, glycosides, terpenoids, alkaloids, and others (Table 2).

### 5.1. Alkaloids

#### 5.1.1. Imperialine (Imp) and Verticinone

Imp is a steroidal alkaloid secondary metabolite isolated from the medicinal herb *Fritillaria imperialis*. Imp has an effect on cell viability, levels of AGEs, MGO, 3-DG formation, GLO-I activity, carbohydrate-hydrolyzing enzymes (α-amylase and α-glucosidase), and glucose uptake capacity [93]. Imp caused a hypoglycemic effect by inhibiting α-amylase and α-glucosidase and significantly inhibited NF-κB phosphorylation and TNF-α and IL-1β production induced by lipopolysaccharide (LPS) in RAW 264.7 macrophages. The anti-inflammatory and antioxidant response of NF-κB plays an important role in wound healing and tissue repair. Moreover, NF-κB activates the migration of cells, proliferation, modulates the expression of matrix metalloproteinases (MMPs), and activates the secretion of growth factors and cytokines for improved wound healing [137]. It was observed that LPS considerably raised the phosphorylation of NF-κB and inhibited Iκ-B phosphorylation which in turn inhibited NF-κB nuclear translocation and transcriptional activity, ultimately reducing the release of inflammatory mediators and cytokines. Additionally, Imp reduces LPS-stimulated NO generation and iNOS protein expression and significantly downregulates the expression of COX-2. NF-κB plays a crucial role in controlling the expression of inflammatory enzymes including iNOS and COX-2 as well as the synthesis of pro-inflammatory cytokines [138]. Verticinone is another active secondary metabolite and an isomer of Imp from *F. imperialis* with a variety of pharmacological effects. Verticinone was shown to have hypoglycemic effects via increasing insulin secretion and glucose absorption. Moreover, it prevents β-TC6 pancreatic and C2C12 skeletal muscle cells from hydrolyzing carbohydrates [139].

#### 5.1.2. Piperine

Piperine, an active component found in both *Piper nigrum* L. and *P. longum*, has a number of biological effects, including anti-inflammatory, immunomodulatory, and antitumor activities. However, the activity of piperine on the structural and functional alterations of glycated proteins as well as the inhibition or reversal of these modifications are not fully understood. As an antidiabetic, it is assumed that it might also have anti-inflammatory and antioxidant properties. Piperine reduced GLO-I activity dose-dependently by 1.31-fold in MCF-7 cells possibly at a post-translational level, since it does not alter GLO-I expression. Additionally, piperine reduced the viability of cells treated with MGO and GO by 1.96 and 1.63 after 24 h and 2.56 and 1.66 times after 48 h, respectively [95]. Piperine has shown an antiglycation effect and prevents the accumulation of AGEs in obesity-related colorectal cancer cells [140]. In cases of hyperglycemia, piperine has inhibited the onset of glycation-induced diabetes consequences in a dose-dependent manner, thereby preventing erythrocyte membrane modifications and oxidative stress against in vitro albumin glycation. It is suggested that piperine consumption on a daily basis can stop the onset of glycation-induced diabetes consequences in hyperglycemic situations [140]. Additionally, studies indicate that piperine can lower blood sugar, triglycerides, and cholesterol levels [141]. Additionally, a recent study demonstrated that piperine increased the TGF-β level, decreased the activation of NF-ĸB, and improved the collagen repair in the periodontal tissues to promote tissue repair [94].

#### 5.1.3. Berberine

Berberine is usually found in the stems, bark, roots, and rhizomes of *Berberis vulgaris* (barberry), *Hydrastis canadensis* (goldenseal), and *B. aristata* (tree turmeric). It has multiple clinical applications in inflammation and chronic disease conditions. It has been demonstrated that berberine slows the progression of prediabetes to diabetes in Zucker diabetic fatty rats by increasing glucagon-like peptide-2 intestinal secretion and enhancing the gut microbiota. Only 30% of rats in the berberine group progressed to T2DM as compared to other groups. Berberine treatment caused decreased food intake, fasting blood glucose (FBG) levels, insulin resistance, and plasma LPS levels as well as raised levels of fasting plasma glucagon-like peptide-2 (GLP-2) and intestinal GLP-2 secretion triggered by glutamine. In impaired glucose tolerance (IGT) rats, berberine was able to reverse the elevated expressions of TLR-4, NF-κB, and TNF-α and the upregulated expressions of mucin, occludin, and ZO-1 [96]. Recently, scientists discovered that berberine reduces insulin resistance and endotoxemia in T2DM patients and rat models of diabetes [96,142,143]. Protoberberine alkaloids berberine and epiberberine isolated from *Coptis chinensis* Franch inhibited the protein tyrosine phosphatase 1B (PTP1B) enzyme, a negative regulator of the insulin signaling pathway and therapeutic target for T2DM [144,145]. Berberine significantly lowers blood glucose levels, prevents AGE formation, increases antioxidant capacity, and protects against diabetic nephropathy through other synergistic pathways [98]. Berberine improved wound closure via the downregulation of MMP9 and upregulation of TGF-β1 in STZ-induced diabetic rats [97]. It has shown improved pancreatic β cell regeneration [146].

### 5.2. Terpenes and Terpenoids

#### 5.2.1. Momordicosides

These are a type of triterpenoid found in *Momordica charantia* (bitter melon) and have many types (such as A, B, C, D, E, G, F1, F2, I, K, and L). About 200 such compounds are known so far. It has shown increased pancreatic insulin production, decreased insulin resistance, increased peripheral and skeletal muscle cell glucose utilization, restricted intestinal glucose absorption, and suppressed key enzymes in the gluconeogenic pathways [103]. Momordicosides have shown antidiabetic properties via different mechanisms [147]. Momordicosides (Q, R, S, U, and T) have been found to be advantageous for diabetic people, including by improving the uptake of inducible glucose into cells and promoting fatty acid oxidation and glucose elimination [148]. AGEs have been found to inhibit GLUT-4 translocation from the cytoplasm to the plasma membrane [149]. However, momordicosides stimulate GLUT4 translocation to the cell membrane, regulate glucose, and have shown antigluconeogenetic activity *via* the activation of AMPK pathway in a mouse model [150].

#### 5.2.2. α-Pinene

Alpha-pinene is found in the oils of many cannabis plants, such as *Cannabis sativa*, *C*. *indica*, and *C. ruderalis,* as well as rosemary and satureja (*Rosmarinus officinalis*, *Satureja myrtifolia*). α-pinene has shown significant hypoglycemic and anti-inflammatory activities in mice [151]. 1S-α-pinene isolated from the extracts of *Foeniculum vulgare* and *Eryngium carlinae* has shown antioxidant and anti-hyperglycemic activity in diabetic rats [99]. 1S-α-pinene from *Ocimum. tenuiflorum* has shown antidiabetic activity via inhibiting DPP4, an enzyme that plays a critical role in glucose homeostasis in diabetes mellitus [152]. α-pinene isolated from the essential oil of *R. officinalis* has shown inhibitory activity against alloxan-induced diabetes and oxidative stress in rat liver and kidney [153]. α-pinene has shown an antihyperglycemic effect by increasing the insulin levels, reducing the plasma glucose level, and helping the cells to better utilize glucose [154]. It has been reported to have creased VEGF expression, which improves wound healing *via* angiogenesis [100].

#### 5.2.3. Stevioside

Stevioside is a diterpene steviol glycoside found in the plant *Stevia rebaudiana*. It is a natural sweetener [155]. It has shown insulinotropic, glucagonostatic, and anti-hyperglycemic properties [156,157]. Stevioside prevents the production of glucose in the liver of diabetic rats. These antihyperglycemic, insulinotropic, and glucagonostatic effects, which are basically plasma glucose level dependent, require high glucose levels. Steviol glycosides influence GLUT4 translocation via the PI3K/Akt pathway to produce their effects [101,158]. The PI3K-AKT pathway plays an important role in diabetic wound healing [100]. Additionally, stevioside inhibits gluconeogenesis in the liver of diabetic rats [101]. Additionally, its analog isosteviol has also shown an anti-T2D and cardioprotective effect by lowering blood glucose and AGEs, inhibiting ERK and NF-κB signaling, and activating endogenous antioxidant mechanisms [102].

### 5.3. Phenols and Polyphenols

#### 5.3.1. Diphlorethohydroxycarmalol (DPHC)

DPHC is a polyphenol isolated from the edible seaweed *Ishige okamurae*. It inhibited MGO- and AGES-induced cytotoxicity and ROS production in the human embryonic kidney cell line (HEK). DPHC upregulated the Nrf2 gene expression with simultaneous upregulation of antioxidant and MGO detoxification enzyme GLO-I. This polyphenol can provide an effective treatment for diabetic nephropathy [104]. It has been observed that DPHC administration in STZ-induced diabetic mice blocked α-glucosidase and α-amylase activities and effectively reduced the rise in postprandial blood glucose levels compared to control mice [159]. DPHC reduced the cellular damage caused by the high glucose-induced oxidative stress associated with diabetes and improved cell viability. DPHC considerably decreased the levels of NO, intracellular ROS, and lipid peroxidation in high glucose conditions in a dose-dependent manner. Moreover, DPHC reduced the high glucose mediated overexpression of the proteins COX-2 and inducible nitric oxide synthase (iNOS), as well as the NF-κB activation in HUVECs [160]. DPHC reduced the glucotoxicity, apoptosis, production of thiobarbituric acid reactive substances (TBARS), intracellular ROS, and elevation of NO levels mediated by high glucose. Additionally, RINm5F pancreatic cells pretreated with high glucose showed enhanced activity of antioxidant enzymes such as catalase (CAT), SOD, and glutathione peroxidase (GSH-px) after DPHC therapy [161].

#### 5.3.2. Resveratrol

Resveratrol is a natural polyphenol found in *Vitis palmata* (red grape), berries, *Arachis hypogaea*, and some vegetables. It has antioxidant properties and has been studied for potential therapeutic use in many disease conditions including inflammation, diabetes, and neurodegeneration [162,163,164]. Resveratrol showed antiglycation ability in a dose-dependent manner, inhibited carbohydrate-hydrolyzing enzyme activity, trapped MGO, and formed resveratrol-MGO adducts [165]. Palsamy et al. (2011) demonstrated that resveratrol has shown a renoprotective property by reducing the oxidative stress and inflammatory cytokines through mediating Nrf2-Keap1 signaling and its downstream regulatory proteins in diabetic rats [166]. Resveratrol increased plasma total antioxidant capacity and reduced plasma protein carbonyl content in T2D patients [167]. Resveratrol has shown beneficial effects on the liver of diabetic rat by reducing MDA levels, total oxidant, plasma glucose, oxidative stress, and reducing the RAGE expression [24]. Resveratrol reduced DM-induced vasculopathy and improved tissue generation through attenuating the RAGE expression and NF-κB signaling pathway [168] (Figure 1).

#### 5.3.3. Caffeic Acid (CA)

CA is found in many plants and foods, such as apples, berries artichoke, and pears. CA reduced intracellular ROS and inhibited AGE formation via the downregulation of IL-1β, IL-18, ICAM-1, VCAM-1, NLRP3, Caspase-1, and CRP (C-reactive protein) in HUVECs exposed to AGEs, suggesting that CA supplementation can be beneficial in delaying AGE-induced vascular dysfunction in diabetes [129]. Additionally, an in silico study by Khan et al. (2022) has revealed that CA can inhibit AGE-induced fluorescence in vitro. A fluorescence quenching study revealed the van der Waals force- and hydrogen bonding-driven formation of the α-amylase-CA complex. It also inhibited protein glycation [169]. CA oligomer prevented AGE formation by inhibiting the increase in 3-DG production in the early phase of the Maillard reaction [170]. CA at a concentration of 5 mM inhibited GO-induced CML formation in the biscuit model system [171]. CA (50 mg kg^−1^) has shown a protective effect against atherogenic outcomes in the liver and kidney of STZ-induced diabetic mice [172].

#### 5.3.4. Vanillin

Natural vanillin, a phenolic aldehyde, is extracted from the seed pods of *Vanilla planifolia* and is used as a flavoring agent in foods and the pharmaceutical and cosmetic industries. It reduced hyperglycemia and enhanced renal function by decreasing renal expression of NF-κB and renal concentrations of cytokines IL-6, TGFβ1, and collagen in a rat model. Additionally, vanillin significantly decreased serum AGES level. It also increased SOD activity and reduced MDA concentration in renal tissues and thereby reduced DN [127]. Vanillin has shown a hypoglycemic effect via anti-AGES activity that protects against nonenzymatic glycation in human insulin and promotes the formation of harmless fibrils [126]. Vanillin treatment significantly lowered fasting blood glucose level in comparison with the DN group in a rat model [127]. Vanillic acid inhibited intracellular glycation systems such as ROS, p38 and JNK, PKC and p47phox, and MGO-derived CML formation to prevent Neuro-2A cell death and DN [128].

### 5.4. Flavonoids

#### 5.4.1. Procyanidins

Procyanidins isolated from cinnamon (*Cinnamomum zeylanicum*) have antioxidant and anti-inflammatory properties and have been shown to inhibit protein glycation effectively. Procyanidins are composed of various ratios of (+)-catechin, (−)-epicatechin, and their derivatives. They can be classified as A-type (connected by C2-O-C7 or C2-O-C5) and B-type (linked by C4-C8 or C4-C6) [173]. Depending on the degree of polymerization, the bonding site, and the steric conformation of the polymer, procyanidins can exhibit variation in their antioxidant activity. Moreover, procyanidins had a substantially lower capability for scavenging GO than MGO due to the high polymerization of GO as a dimer and trimer in aqueous solution and less availability of free GO [11]. The major mechanism of the antiglycation of carbonyl scavenging property is MGO scavenging [125]. Oligomeric procyanidins, due to their anti-inflammatory and insulin-potentiating properties, are useful in reducing the symptoms of diabetes and Alzheimer’s disease.

#### 5.4.2. (+)-Catechin

Catechin is a flavan-3-ol isolated from green tea, black tea, fruits, and cacao products. Catechin could inhibit α-amylase, β-glucosidase, and α-glucosidase and mediate MGO trapping that reduced AGE formation [91]. It lowered proinflammatory cytokines TNFα and IL-1β in an MGO-mediated DN model [106]. It was observed that catechin was effective in inhibiting the production of ROS, which may be the main factor in inhibiting the AGE and CML formation. Moreover, molecular docking research also revealed a better inhibitory effect of catechin on β-glucosidase [174]. It shows scavenging reactive carbonyls by forming CC-MGO adducts and CC-GO adducts. The structure–activity link between the catechin ligand and enzymes has been established by molecular docking experiments that examined the various catechin ligand interactions within the active site pockets of α-amylase and β-glucosidase [174]. It activated muscle stem cells (MSCs) and increased muscle regeneration via Myf5 induction in a C57BL/6 mice model [175].

#### 5.4.3. Chalcones (1,3-Diaryl-2-Propen-1-Ones)

Chalcones are found in the yellow flower pigments of Coreopsis and other Asteraceae (Compositae) species, Leguminosae family (Fabaceae), Cannabaceae, Piperaceae, Solanaceae, Anacardiaceae, and Caesalpiniaceae. Chalcone (CHA79) reduced the MGO-induced GLO-I activity. It has demonstrated neuroprotective effects against MGO med-ated damage via enhancing neurotrophic signal, antioxidant defense, and the anti-apoptosis pathway. It has increased the expression of neurotrophic factors such as p75NTR, p-TrkB, p-Akt, p-GK-3β, and p-CREB as well as the glucagon-like peptide-1 receptor (GLP-1R) and brain-derived neurotrophic factor (BDNF) [107]. Therefore, it can be useful in formulating new medications to treat PD or other neurodegenerative illnesses.

#### 5.4.4. Tectorigenin

Tectorigenin is a natural isoflavonoid isolated from the *Pueraria thomsonii* Benth and *Belamcanda chinensis*. It can effectively regulate insulin action in the endothelium by inhibiting palmitic acid-induced mitochondrial membrane damage as well as ROS production and inflammation by the downregulation of IKKβ/NF-κB phosphorylation and JNK activation that resulted in reduced TNF-α and IL-6 production in endothelial cells. Moreover, it can restore impaired insulin PI3K signaling and inhibit inflammation-induced IRS-1 serine phosphorylation, which reduces NO generation. Tectorigenin effectively restored the loss of insulin-mediated vasodilation in the rat aorta by reducing endothelin-1 and vascular cell adhesion molecule-1 overexpression, suggesting a potential application in treating diabetes-associated cardiovascular diseases (CVDs) [176]. Tectorigenin significantly reduced serum glucose by 53% in STZ-induced hyperglycemic SD rats. Moreover, tectorigenin reduced serum cholesterol levels by 61% [177]. Tectorigenin inhibited renal inflammation and DN *via* reducing macrophage infiltration and restored the reduced expression of adiponectin receptor 1/2 (AdipoR1/2) in db/db mice, suggesting a potential role of AdipoR1/2 in DN treatment [116]. Tectorigenin significantly improved the viability of pancreatic β-cells in a hyperglycemic condition by activating the ERK pathway. It also decreased islet β-cell apoptosis in the pancreas of mice fed a high-fat/high-sucrose diet (HFHSD), thereby significantly preserving or regaining islet size and β-cell mass, respectively, by upregulating pancreas/duodenum homeobox protein 1 (PDX1) expression. The amelioration of hyperglycemia and glucose intolerance showed that tectorigenin was effective both as a preventative measure and as a treatment for HFHSD-impaired glucose metabolism in mice [117].

### 5.5. Tannins

Tannins are high-molecular-weight polyphenolic compounds found in coffee, strawberries, tea, clove, wine, dry fruits, and grapes and in a variety of plants including oak wood (*Castanea sativa*, *Quercus petraea*, *Caesalpinia spinosa*, *Rhus coriaria*, *Terminalia chebula* and *Q. infectoria*). In nature, there are two types of tannins, i.e., condensed and hydrolysable. Hydrolysable tannins include gallic acid and ellagic acid, whereas condensed tannins include proanthocyanidins, procyanidin B2, gallocathecin, and epigallocathecin [178]. Tannins can bind efficiently to proteins and other biomolecules including amino acids and alkaloids and make them precipitate. Due to the health benefits associated with their antioxidant and hypoglycemic properties, tannins can be potential compounds in the prevention and management of diabetes and the associated complications [179]. Due to their antihyperalgesic, anti-inflammatory, and antioxidant properties, tannins may be useful in the treatment of DN [180].

#### 5.5.1. Gallic Acid (GA)

GA is a trihydroxybenzoic acid found in a variety of edible plants, including oak bark, *Caesalpinia mimosoides*, *Cynomorium coccineum*, tea leaves, sumac, and witch hazel. It is used as an antioxidant and immunity booster due to its capacity to scavenge free radicals, which helps to prevent or treat oxidative stress, a major factor in DM and the related complications [181]. GA improved MGO-induced DN via a decreased Nrf2 level and elevated GLO-I activity as well as improved infiltration of inflammatory cells and morphology of proximal epithelial cells of mice kidney. GLO-I detoxifies MGO, which ultimately reduces the production of AGEs in the kidneys. Nrf2 is a critical activator of antioxidant defense that traps MGO through enhanced GLO-I activity. GA showed an anti-fibrogenic effect in MGO-treated mice kidney by downregulating miR-192 and upregulating miR-29a. However, GA-induced miR-204 upregulation indirectly regulates the Nrf2 level. Moreover, GA decreased oxidative stress in MGO-treated mice kidney via decreased MDA and increased GSH, CAT, and SOD activity [115]. GA reduced the effects of MGO-induced diabetes on the reproductive system by reducing oxidative stress and free radical generation in a mice model [182]. GA reduced the elevated circulatory pro- and anti-inflammatory cytokines, chemokines, N-εCML, CRP, and glycosylated HbA1c levels in dAGE-fed mice [183]. The expression of inflammatory cytokines NOX and RAGE as well as abnormal matrix protein expressions was significantly reduced in GA pre-treated H9C2 (2–1) cells [184].

#### 5.5.2. Gallotannins

Glucitol-core containing gallotannins (GCGs) have been isolated from red maple (*Acer rubrum*) and sugar maple (*A. saccharum*). GCGs have shown anti-inflammatory, anti-glucosidase, and anti-diabetic properties. GCGs can effectively chelate ferrous iron, an oxidative catalyst of AGEs formation that supports the antioxidant mechanism of the antiglycating activity of GCGs [185]. Penta-O-galloyl-d-glucopyranoside (PGG), a hydrolyzable derivative of gallotannin, prevents AGEs formation by protecting protein structure via reducing the transition from α-helix to β-sheets by up to 50% in BSA [186]. Gallotannin has a protective effect on cell death signaling because it is a poly(ADP-ribose) glycohydrolase (PARG) inhibitor and also protects against Poly(ADP-ribose) polymerase (PARP) breakage, a hallmark of apoptotic cell death. Treatment with gallotannin (20 mg/kg/day, i.p.) in the STZ-induced model of diabetes for 4 weeks resulted in significantly reduced plasma creatinine, which is an important marker for DN [114].

#### 5.5.3. (−)-Epigallocatechin-3-O-Gallate (EGCG)

In STZ-induced diabetic rats, treatment with EGCG reduced the hyperalgesia reactions seen in the hot plate, tail immersion, formalin, and carrageenan-induced oedema tests. In another study, the paw withdrawal threshold (PWT) of STZ-induced diabetic rats who had received EGCG treatment for 10 weeks reduced the mechanical hyperalgesia and tactile allodynia. EGCG orally at a dose of 25 mg/kg for 5 weeks can decrease serum blood glucose levels, enhance serum lipid profiles, and boost body weight via reducing inflammatory biomarkers such as IL-6, NO, and TNF-α [187]. In STZ-induced diabetic rats, EGCG (25 mg/kg and 50 mg/kg orally for 28 days) decreased MDA but increased the decreased GSH, catalase, and SOD levels [188]. Administration of EGCG (20 and 40 mg/kg BW) orally reduced blood MDA and NO levels while raising SOD levels in male albino Wistar rats. Chronic EGCG therapy (40 mg/kg) significantly reduced hyperalgesia in diabetic rats compared to those who were not receiving treatment [113].

#### 5.5.4. Proanthocyanidins and Procyanidin B2

In a study conducted on male Wistar rats, Cui et al. (2008) discovered that proanthocyanidins isolated from grape seed might increase body weight while lowering AGEs and HbA1c, but not blood glucose levels. A 24-week treatment of proanthocyanidins at 250 mg/kg/daily (intragastric) decreased plasma MDA and elevated SOD levels [189]. Proanthocyanidin B2 (10 g/mL) treatment in high-glucose dorsal root ganglia (DRG) culture reversed the neurotoxic effect brought on by the glucose challenge [110]. High glucose concentration affects the primary afferent neurons in the DRG. This glucose challenge results in hyperglycemia, which prevented neuronal growth, leads to oxidative stress and mitochondrial dysfunction, and resulted in apoptotic cell death in DRG [111,112].

### 5.6. Saponins

Saponins are glycosides that have a wide range of biological and pharmacological effects due to the great diversity of structures they express on both sugar chains and aglycones or sapogenin. An aglycone unit is combined with one or more carbohydrate chains to form saponins. The aglycone or sapogenin unit is made up of a sterol or the more prevalent triterpene unit. These glycosidic compounds, often referred to as sterol saponins, have been shown to have potential therapeutic effects and are being investigated as a possible substitute for insulin in diabetic patients. These are widely distributed in nature, and the most common saponin-rich plants are soapwort, soybean (*Glycine max* L.), and soapbark tree *Quillaja saponaria*. They are one of the main active ingredients in traditional Chinese medicine as well as other folk remedies. Due to the microheterogeneity of saponins, isolating them from natural sources is typically a difficult operation. Saponins are a structurally varied class of chemicals with a glycosyl residue-attached skeleton produced from the 30-carbon precursor oxidosqualene. There are different kinds of saponins depending on the type of carbon skeleton. Saponins have a complex structure due to variations in the aglycone structure, type of side chain(s), and the positions at which these moieties are attached to the aglycone [190]. These carbon skeletons can undergo different chemical reactions such as homologation, fragmentation, and degradation.

Saponins have been recognized as an antidiabetic principle from medicinal plants, and numerous reports on their antidiabetic benefits have been published. The antidiabetic effects of saponins have been attributed to a number of different mechanisms, including the activation of PPAR gamma (Peroxisome proliferator-activated receptors gamma), activation of GLUT4, activation of adiponectin expression, activation of the PI3K/Akt pathway, increased expression of adipsin, and activation of the AMP-activated protein (AMPK) [119,120,191,192].

#### 5.6.1. Kaikasaponin III (KS III)

KS III significantly reduced serum glucose by 39%, inhibited body weight loss by 48% in a 10 mg/kg dose and 36% in a 5 mg/kg dose as well as reduced serum cholesterol levels by 20% in STZ-induced hyperglycemic SD rats [177]. KS-III has shown hypoglycemic and hypolipidemic effects. In STZ-treated rats, KS-III boosted tissue factor activity and lengthened the bleeding and plasma clotting times, indicating that this substance has anti-thrombotic effects. Additionally, it reduced the production of MDA and hydroxy radicals in the liver and serum while enhancing SOD activity. SOD, glutathione peroxidase, and catalase activity increased in response to KS-III, indicating the activation of free radical-scavenging enzymes [123]. In a rat model, the anti-lipid peroxidative effect of KS III demonstrated a significant reduction in MDA formation [124].

#### 5.6.2. Astragaloside IV

Astragaloside IV is a glycoside of cycloartane-type triterpene discovered in *Astragalus membranaceus*. In STZ-induced diabetic rats, it has shown a protective effect against the development of peripheral neuropathy. It has a hypoglycemic effect on blood glucose levels and increases plasma insulin levels. Another study found that astragaloside IV at doses of 25 and 50 mg/kg significantly reduced blood glucose, triglyceride, and insulin levels. Moreover, it suppressed the expression of glycogen phosphorylase and glucose 6 phosphatase in diabetic mice [121,122,193].

#### 5.6.3. Diosgenin

Natural diosgenin [25R-spriost-5-en-3β-ol], a steroidal saponin isolated from the rootstock of yam, has several pharmacological activities, including a hypoglycemic effect. It reduces intestinal sucrose levels in diabetic male Wistar rats. It also increases glucose-6-phosphate activity [194,195]. Diosgenin is effective in the management of diabetes-related hepatic dyslipidemias by decreasing hepatic triglyceride content in diabetic mice and inhibiting LXRα activity in HepG2 cells in hyperglycemic conditions [196]. Diosgenin has shown potential effects on hyperglycemia-associated cardiovascular risk, insulin secretion, and beta cell regeneration in STZ-induced diabetic rats. In this study, after 30 days of treatment with diosgenin, fasting blood glucose, glucose-6-phosphatase, fructose-1,6-bisphosphatase, and LPO levels were significantly increased, whereas HDL, SOD, CAT, GSH, and the glycolytic enzyme glucokinase levels were significantly decreased [118] (Figure 2).

#### 5.6.4. Platyconic Acid (PA)

PA (2”-O-acetyl platyconic acid A) has shown hypoglycemic activity and can efficiently decrease blood glucose level and increase insulin activity [197]. It efficiently increased insulin-stimulated glucose uptake in 3T3-L1 adipocytes, presumably in part through acting as a peroxisome proliferator-activated receptor (PPAR-activator). During the initial stage of oral glucose tolerance tests (OGTT), diabetic mice treated with PA showed the lowest peak serum glucose and highest serum insulin levels. Adiponectin and PPAR expression in adipose tissue were potentiated by PA, which also improved insulin signaling and increased GLUT4 translocation into the membranes. PA also increased glycogen accumulation and decreased triacylglycerol storage in the liver, which was related to enhanced hepatic insulin signaling [192]. According to recent research, PA improves insulin sensitivity, promotes the storage of glycogen, and speeds up the translocation of the glucose transporter GLUT4 into membranes [119].

### 5.7. Phytosterols

Plant sterols, or phytosterols, are a group of compounds structurally similar to mammalian cell-derived cholesterol and are an essential structural element of cell membranes. They occur naturally in a wide range of plants, including soybean, rapeseed, olive, walnut, grapeseed, and peony seed oils. The use of phytosterols in the diet may lower the risk of coronary heart disease in people. Plant sterols have a double bond in the sterol ring. The most abundant sterols in plants and the human diet are β-sitosterol, campesterol, and stigmasterol.

#### 5.7.1. β-Sitosterol

Sitosterol is used to treat hypercholesterolemia, coronary artery disease, and prostate and breast cancer. After giving a high-fat diet to STZ-induced diabetic rats and β-sitosterol at a dose of 15 mg/kg body weight per day for 30 days, the levels of plasma glucose and glycosylated hemoglobin and the homeostatic model assessment of insulin resistance significantly decreased, while the levels of insulin, Hb, and the protein expression of PPAR and GLUT4 in tissues that depend on insulin increased [130,198]. It decreased glycated Hb, NO, and serum glucose with a concomitant increase in serum insulin levels in STZ-induced diabetic rat models [199]. In a study on a diabetic rat model, it was observed that β-sitosterol reduces insulin resistance in adipose tissue through IRS-1/Akt-mediated insulin signaling. β-sitosterol significantly decreases high-fat diet-induced serine phosphorylation of insulin receptor substrate IRS-1 and increased the tyrosine phosphorylation of IRS-1 (p-IRS-1 Tyr632), serine phosphorylation of Akt (p-AktSer473), threonine phosphorylation of Akt substrate of 160 KD (p-AS160 Thr642), and threonine phosphorylation of Akt (p-AktThr308) in the adipose tissue of diabetic rats [200]. Through the downregulation of the IKKβ/NF-B and c-Jun-N-terminal kinase (JNK) signaling pathway, β-sitosterol reduces the inflammatory events in adipose tissue, which in turn prevents obesity-induced insulin resistance in diabetic rat models [201].

#### 5.7.2. Stigmasterol

Stigmasterol has anti-amylolytic properties, delaying glucose release and reducing hyperglycemic effects [131]. It has shown antiglycation activities that reduced AGEs formation [202]. Stigmasterol has shown an antihyperglycemic effect in an alloxan-induced diabetic rat model. Diabetic rats administered with stigmasterol (100 mg/kg b. wt.) for 4 weeks showed reduced HbA1c and fasting blood glucose levels and reversed the serum/urine protein, urea, and creatinine abnormalities significantly. It also reduced diabetes related symptoms such as urine glucose, polyuria, and polydipsia. In addition, stigmasterol treatment caused glucose-metabolizing enzymes, including pyruvate kinase, lactate dehydrogenase, fructose-1,6-bisphosphatase, glucose-6-phosphate dehydrogenase, and hexokinase to revert to their normal levels. It also improved the regeneration of pancreatic beta cells [132]. In another study, stigmasterol has shown a hypoglycemic effect via an increased expression and translocation of the GLUT4 transporter in L6 cells, white adipose tissue, and skeletal muscle [203]. Stigmasterol has anti-atherosclerotic properties, which reduced the glucolipotoxicity-mediated rise in free cholesterol and ROS levels in INS-1 cells. Moreover, adding stigmasterol to cells exposed to glucolipotoxicity prevented early apoptosis, increased total insulin, and enhanced actin rearrangement and insulin production [204].

### 5.8. Natural Peptides

Carnosine (β-alanyl-L-histidine), a natural peptide, is widespread in vertebrate tissues. It possesses a variety of biochemical properties, including antioxidant, bivalent metal ion chelation, proton buffering, and carbonyl scavenger activities [133]. It is absorbed into the plasma unaltered and intact and inhibits the production of AGEs by lowering blood sugar, preventing early glycation, and even reversing already created AGEs. Aside from these, glutathione, homocarnosine, homoalanine, and L-histidine are other interesting natural peptides and amino acids. Foods containing bioactive peptides and amino acids may be used to enhance health if they can reduce the production of AGEs [134].

Another important natural peptide, polypeptide-p, has been found in the fruits, callus, and seeds of *M. charantia* (MC), which contains insulin-like proteins [205]. According to Indian Ayurveda, MC was regarded as a remedy for diabetes mellitus in India for thousands of years. These proteins are homologous to human insulin, and have shown hypoglycemic effect on rats, langurs, gerbils, and human [135,136,148,206].

### 5.9. Manuka Honey (MH)

The chief natural source of MH is the *Leptospermum scoparium* tree of the Myrtaceae family. Due to its phytochemical profile and higher nutritional value compared to other types of honey, it is becoming a more popular functional food. It contains a variety of proteins, amino acids, flavonoids such as galangin and pinocembrin, antibiotic-rich inhibine, and phenol antioxidants. Research reveals that it has several beneficial biological properties, including immune-stimulatory, wound-healing, antibacterial, antiviral, antioxidant, and anti-inflammatory effects [207]. The antioxidant properties of honey reduce the formation of advanced glycation and lipoxidation end products in diabetics [208]. The antioxidant, anti-inflammatory, enhanced wound healing, and antibacterial activity of MH is due to its MGO content [209,210,211]. The non-peroxide bacteriostatic activities of MH are linked to MGO. The GO and MGO of MH have immunomodulatory effects that can help improve tissue regeneration and wound healing [212]. The clinical uses of MH have been focused on tissue regeneration and wound healing. It is being used in the field of tissue engineering to create a template for regeneration. MH reduced oedema and leukocyte infiltration in a mouse model and also reduced neutrophil superoxide generation in vitro [213]. Nevertheless, the study was unable to identify the precise component causing the inflammatory effect. The TLR1/TLR2 signaling pathway directly linked the phenolic content of MH to its anti-inflammatory effect. Sell et al. (2012) has demonstrated that its use can accelerate tissue regeneration in wound healing [214]. The underlying mechanism includes the stimulation of cytokines that cause enhanced tissue growth via the proliferation of fibroblasts and macrophage infiltration at the wound site when a wound is dressed with MH. Chrysin from honey can suppress the activity of nitric oxide synthase (iNOs), COX-2, and proinflammatory enzymes [215]. Another study showed that MH incorporated into a composite dressing with poly(vinyl alcohol) (PVA) has improved wound healing properties [216]. MH-impregnated dressings reduced the time for wound healing in patients suffering from neuropathic diabetic foot ulcers [217]. Research shows that MH activated both the anti-inflammatory cytokine IL-6 and the proinflammatory cytokines TNF-α and IL-1β in monocytes [218]. Due to its unique chemical composition and biological properties, the development of medical-grade honey is an interesting subject among various clinical and biomedical engineering groups that can transform the medical industry.

## 6. Conclusions and Future Directions

In a cellular system, the activation of the AGE-RAGE signals induce oxidative stress and inflammatory responses that endorse AGEs-related disease conditions. Hyperglycemia-induced inflammatory responses damage vascular endothelial, renal mesangial, smooth muscle, and nerve cells. Moreover, environmental factors, such as high dAGEs content, smoking, and other factors, enhance AGEs formation in addition to hyperglycemia. Due to GLO-I insufficiency, patients with renal proximal tubular injury delay the breakdown and removal of intracellular AGEs. Phytochemicals, functioning as AGEs formation inhibitors, preformed AGEs breakers, AGE-RAGE axis blockers, and GLO-I stimulators, can provide potential solutions to diabetes-associated conditions including CVDs, nephropathy, delayed wound healing, impaired regeneration, and neuropathy. The MGO-trapping capacity of certain phytoconstituents can prevent AGEs accumulation and related illness. Molecular pathways such as the inhibition of NF-κB phosphorylation and NF-κB nuclear translocation, upregulation of TGF-β, TLR-4, TNF-α, ZO-1, GLP-2, and TGF-β1 by phytoconstituents can reduce AGEs and hyperglycemia-related complications. The upregulation of VEGF expression by phytoconstituents may improve the angiogenesis process. Moreover, enhanced GLUT4 translocation, PI3K-AKT, and downregulation of ERK may improve impaired tissue regeneration and thereby enhance diabetic wound healing. Enhanced neurotrophic signals due to reduced antioxidant defense, anti-apoptosis pathway, and reduced GLO-I activity can be reverted efficiently by certain flavonoids. The activation of antioxidant pathways results in lowered AGEs, HbA1c, and MDA, which reduce DN. miR-204 upregulation, reduced NOX, RAGE, MMPs expression, downregulation of IKKβ/NF-κB phosphorylation, Nrf2, AdipoR1/2, JNK activation, TNF-α and IL-6, reduced macrophage infiltration, and ERK and PDX1 activation may reduce diabetes-associated CVDs and diabetic nephropathy.

The molecular structure of phytoconstituents plays a crucial role in manifesting their biological activity. The continual investigation of natural compounds and their structural requirements is an important factor in hastening the discovery and application of new inhibitors in the food and pharmaceutical industries. Interestingly, sesquiterpenoids are more effective in diabetes due to their compact molecular structure, minimal ramification, and lack of symmetry [219]. Flavonoids can conjugate with active dicarbonyl compounds at the A ring, which is their active site. The molecular weight and monosaccharide composition of polysaccharides are related to their ability to reduce AGE levels. However, there is a dearth of systematic data about the relationship between the structure and inhibitory efficacy. The triterpenes that contain hydroxyl and carboxyl groups have significant potential for treating diabetes via different signaling pathways [220]. Additionally, terpenes, due to the presence of more hydrogen bonds and hydrophobic interactions, have shown an increased anti-diabetic effect [221]. Due to exceptional economic interest, the pharmaceutical and medical industries are curious about finding potential functional components from natural sources for the development of phytoconstituent-based drugs or medical food. However, plant-based functional foods are produced from a mixture of different phytoconstituents rather than from a single ingredient. The combination of phytoconstituents in whole foods with a range of biological functions may have additive or synergistic effects. Among all, flavonoids and polyphenols have been found to have excellent antioxidant activation action with no or least toxicity. Clinical trials to examine the pharmacological effects and well-designed trials in humans are required for the development of phytoconstituent-based medical foods. However, the absorption and bioavailability of phytoconstituents in the cellular system is usually low, which may limit their uses as functional drugs or food.

## Figures and Tables

**Figure 1 ijms-24-07672-f001:**
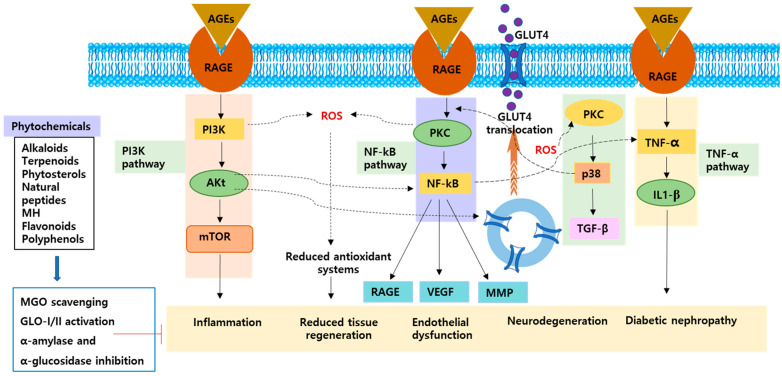
Inhibition of AGE-induced disease conditions by phytoconstituents. AGEs induce different molecular pathways leading to oxidative stress, inflammation, reduced tissue regeneration, ED, DN, and neurodegeneration. Phytochemicals inhibit or prevent these adverse effects by entrapping or scavenging MGO intracellularly. PI3K, NF-kB, and TNF-α signaling pathways are most commonly affected by AGE-induced ROS generation and oxidative stress that can be ameliorated by potential phytoconstituents.

**Figure 2 ijms-24-07672-f002:**
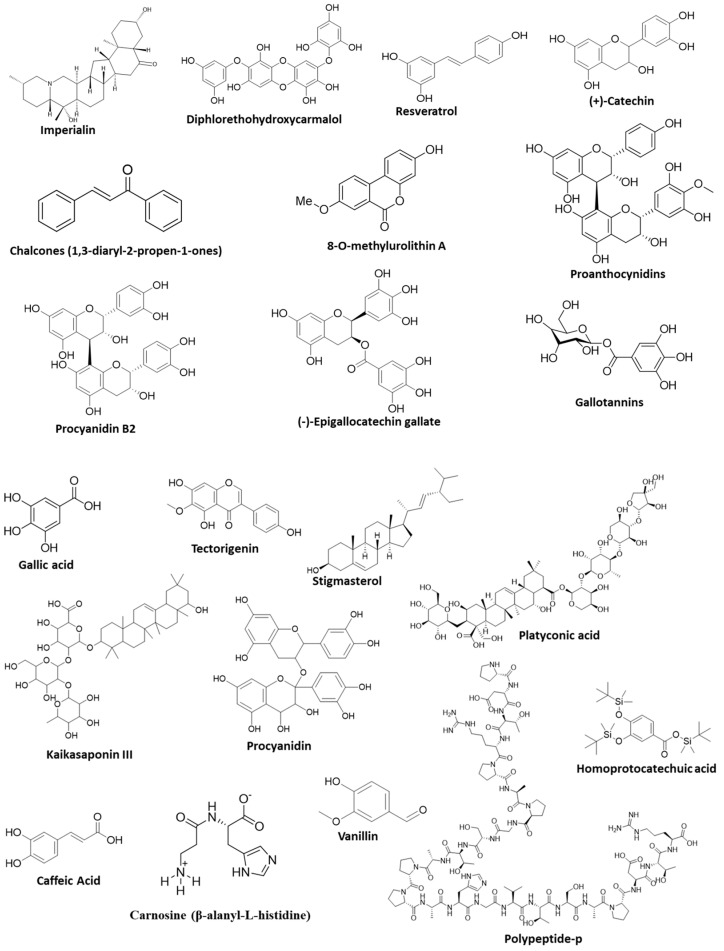
Chemical structures of potential phytoconstituents of pharmacological importance in antiglycation and antihyperglycemic effects.

**Table 1 ijms-24-07672-t001:** The most frequent AGEs in the hyperglycemic condition. AGEs are produced or accumulated in different body parts at different rates and concentrations which augment different pathological conditions via induction of different molecular pathways and transcription factors.

AGE	Body Part Affected	Tissue/Cell Type Affected	Pathological Condition	Mechanisms	Transcription Factor/Mechanism Involved	References
Nonfluorescent crosslinked: Glyoxal-lysine dimer (GOLD)	Kidneys	Mesangial cells	Nephropathy	Free radical generation, inflammation, oxidative stress, mitochondrial dysfunction, NADPH oxidase activity	NRF2	[24,25,26]
Methylglyoxal-lysine dimer (MOLD)	Kidneys	Mesangial cells	Inflammation-related diabetic and nondiabetic renal diseases	Inflammatory cytokine production, ROS production, and mitochondrial dysfunction	NF-κB, PI3K/AKT	[27,28]
3-deoxyglucosone dimer (DOLD)	Brain, kidney, heart	Cerebrospinal fluid, proximal tubule cells’ endolysosome system	AD, cognitive impairment, CKD, heart dysfunction	Increased production of DOLD in cerebrospinal fluid of subjects with AD, neuroinflammation, oxidative stress, upregulation of vascular cell adhesion molecule-1 (VCAM-1) expression, downregulated surface Na+/K+ATPase pumps	NF-κB, angiotensin II, VCAM-1	[29,30,31,32]
Imidazolium crosslink derived from methylglyoxal and lysine-lysine (MODIC)	Skin	ECM, endothelial cells, collagen	Aging, anoikis, impaired angiogenesis, impaired attachment of endothelial cells to type IV collagen, reduced cell binding	Increased cell death in ECM, increased fragility of ECM proteins	α1β1 and α2β1 integrins	[33]
Fluorescent and crosslinked: Pentosidine	Kidneys, heart, brain	Renal cells, heart muscles and cells, serum albumin, neurons	Impaired renal function, CKD, CVD, CAD, AD, increased mortality risk, fluid overload, malnutrition, serum creatinine, increased HbA1C	Increased pro-oxidative state, inflammation, oxidative stress, impaired renal infiltration	hsCRP, IL-6, 8-OHdG	[34,35]
Vesperlysine A, C	Eye	Extracellular matrix1 (ECM1) of human lens, collagen	Lowered vision, aging of eye in diabetics	Increased collagen crosslinking, lens crystalline pigmentation	Collagen	[36,37]
Fluorescent non-crosslinked: Argpyrimidine	Bronchi, blood, human lenses, kidney, brain	Airway epithelial cells, erythrocyte, brunescent cataractous lens proteins, arterial wall of kidneys of diabetic patients, familial amyloidotic polyneuropathy	Refractory schizophrenia, CKD, cataract, lens aging, familial amyloidotic polyneuropathy	Increased immunoreactivity of lens proteins, cataractogenesis, formation of β-fibrils in amyloid deposits, β-fibrils in serum albumin, neuronal inflammatory	Brunescent cataractous lens proteins, human lens α-crystallin, TNFα, NF-κB, Hsp70, JNK	[38,39,40,41,42]
Nonfluorescent and non-crosslinked: CEL, CML, Pyrraline, Pyrraline immine	Blood, brain	Serum, microglial cells, neurons	Aging, AD	Increased ROS production, oxidative stress, mitochondrial dysfunction, reduced cellular ATP reservoir	Mitochondrial-mediated pathway, oxidative pathway	[43]

**Table 2 ijms-24-07672-t002:** Potential phytoconstituents from natural sources that exhibit antiglucotoxicity and antiglycation effects through regulation of various signaling pathways.

Compound	Class	Origin	MGO Targeted Pathway	Disease	Pharmacological Effect	References
Imperialine	Steroidal alkaloid	*Fritillaria imperialis*	Inhibiting α-amylase and α-glucosidase	Diabetes	Hypoglycemic effect, inhibition of NF-κB nuclear translocation,	[93]
Piperine	Alkaloid	*Piper nigrum* L. and *Piper longum*	Antiglycation, reduced GLO-I activity	Diabetes and associated cardiovascular complications	Upregulation of TGF-β, downregulation of NF-κB	[94,95]
Berberine	Alkaloid	*Berberis* *vulgaris*	Prevents AGEs formation, increases antioxidant capacity, increasing glucagon-like peptide-2 intestinal secretion	DN	Upregulation of TLR-4, NF-κB, TNF-α, mucin, occludin, ZO-1, GLP-2, and TGF-β1, downregulation of MMPs	[96,97,98]
α-Pinene	Terpene	*Cannabis* *sativa*	Hypoglycemic and anti-inflammatory	Diabetes	Upregulation of VEGF expression, improved angiogenesis	[99,100]
Stevioside	Diterpene steviol glycoside	*Stevia rebaudiana*	Reduced AGEs formation, anti-hyperglycemic	Diabetes-induced delayed wound healing	Enhanced GLUT4 translocation, PI3K-AKT, downregulation of ERK and NF-κB, improved diabetic wound healing	[101,102]
Momordicosides	Terpenoids (triterpenoid)	* Momordica charantia*	Gluconeogenic pathways	Diabetes	Increase pancreatic insulin production, decrease insulin resistance, increase peripheral and skeletal muscle cell glucose utilization, restrict intestinal glucose absorption	[103]
Diphlorethohydroxycarmalol (DPHC)	Polyphenols	*Ishige* *okamurae*	Inhibition of reducing MGO-induced AGEs, MGO induced cytotoxicity, and ROS production, Nrf2 gene expression, upregulation of Glyoxalase-1	CKD	Reduction of MGO-induced AGEs formation	[104]
Resveratrol	Polyphenolic flavonoid	Citrus fruits, apples, onions, parsley, sage, tea, and red wine, olive oil, grapes, dark cherries, and dark berries such as blueberries, blackberries, and bilberries	Antiglycation mechanism, antioxidant mechanism	Diabetes and associated diseases	MGO trapping capacity, preventing AGEs accumulation	[105]
(+)-Catechin	Flavan-3-ol	Green tea, black tea, fruits and cacao products	TNFα and IL-1β	DN	MGO trapping, reduced AGE formation,	[106]
Chalcones (1,3-diaryl-2-propen-1-ones)	Flavanoid	*Coreopsis*	p75NTR, p-TrkB, p-Akt, p-GK-3β, and p-CREB, glucagon-like peptide-1 receptor (GLP-1R), BDNF	Diabetic neuropathy	Reduced GLO-1 activity, enhancing neurotrophic signal, antioxidant defense, and anti-apoptosis pathway	[107]
8-O-methylurolithin A	Phenol	Cereals, coffee beans, fruits, olives, vegetables, and tea leaves	Anti-inflammatory, antioxidant, hypoglycemic	Hyperglycemia	AGEs inhibition	[108]
α-Glucosidase	Coumarin	Tonka beans, liquorice, and cassia cinnamon	Hypoglycemic	Hyperglycemia	AGEs inhibition	[109]
Proanthocynidins	Tannin	*Quercus infectoria,* *Caesalpinia spinosa*	Antioxidant system activation	Diabetic neuropathy	Lowering AGEs, HbA1c, MDA	[110]
Procynidin B2	Tannin	*Cinchona pubescens*	Hypoglycemic, antioxidant system activation	Dorsal root ganglia	Respiratory chain activation, mitochondrial proteome expression, neuronal growth activation	[111,112]
(−)-Epigallocatechin-3-O-gallate	Tannin	*Camellia sinensis*	Hypoglycemic, anti-inflammatory, antioxidant system activation	Diabetic hyperalgesia	Lowering MDA	[113]
Gallotannins	Tannin	*Acer rubrum*	Ferrous ion chelation	DN	Inhibition of AGEs formation, inhibition of transition from α-helix to β-sheets of proteins	[114]
Gallic acid	Tannin	*Caesalpinia mimosoides,* *Cynomorium coccineum*	Decreased MDA and increased GSH, CAT, SOD activity, GLO-I activation, reduced ROS	DN	Nrf2 downregulation, miR-204 upregulation, reduced NOX, RAGE, MMP expression	[115]
Tectorigenin	Isoflavonoid	*Pueraria thomsonii,* *Belamcanda chinensis*	Restoration of impaired insulin, anti-inflammatory activation, hypoglycemic	Diabetes- associated CVDs, DN	Downregulation of IKKβ/NF-κB phosphorylation, AdipoR1/2, JNK activation, TNF-α and IL-6, reduced macrophages infiltration, ERK and PDX1 activation	[116,117]
Chalcones (1,3-diaryl-2-propen-1-ones)	Flavonoid	Coreopsis, Cannabaceae, Piperaceae	Antioxidant system activation, MGO scavenging, GLO-I activation	Diabetic neurodegeneration	Upregulation of p75NTR, p-TrkB, p-Akt, p-GK-3β, p-CREB, GLP-1R, BDNF	[107]
Homoprotocatechuic acid	Phenol	*Olea europaea*	Antiglycoxidative	Hyperglycemia	Oxidation protein products (AOPPs) and AGEs inhibition,	[108]
Diosgenin	Steroid	Rootstock of yam	Hypoglycemic	Hyperglycemia Related cardiovascular diseases	Decrease glycolytic enzyme glucokinase levels, increase glucose-6-phosphatase and fructose-1,6-bisphosphatase	[118]
Platyconic acid	Saponin	*Platycodon grandiflorum*	Hypoglycemic, improved insulin secretion	Diabetes-induced glycogen storage, impaired hepatic insulin	PPAR activation, increased GLUT4 translocation	[119,120]
Astragaloside IV	Triterpene	*Astragalus membranaceus*	Hypoglycemic, improved plasma insulin, decreased triglyceride, hypoglycemic, improved plasma clotting	Diabetes, peripheral neuropathy	Downregulation of glycogen phosphorylase and glucose 6 phosphatase	[121,122]
Kaikasaponin III	Saponin	*Pueraria montana*	Reduced MDA and OH•, antioxidant system activation	Diabetes, delayed wound healing	Fibrinogen activation, activation of SOD, GSH-Px, and catalase	[123,124]
Procyanidins	*Flavonoid*	*Cinnamomum zeylanicum*	Antioxidant system activation, anti-inflammatory	Diabetic neuropathy, AD	Lowering protein glycation, scavenging GO and MGO	[125]
Vanillin	Phenolic aldehyde	*Vanilla planifolia*	Antiglycation, antioxidant system activation, hypoglycemic, increased SOD activity and reduced MDA	DN	Upregulation of IL-6, TGFβ1, and collagen, p38 and JNK, PKC and p47phox expression	[126,127,128]
Caffeic Acid (CA)	Terpenoid	*Eucalyptus* *globulus*	Antiglycation, antioxidant system activation, 3-DG inhibition	Diabetes-associated vascular dysfunction	Downregulation of IL-1β, IL-18, ICAM-1, VCAM-1, NLRP3, Caspase-1, and CRP	[129]
β-sitosterol	Steroid	Rice bran, wheat germ, corn oils peanuts, and soybeans	Antioxidant and antidiabetic	Diabetes	Upregulation of peroxisome proliferator-activated receptor γ (PPARγ) and GLUT4	[130]
Stigmasterol	3β-sterol	*Glycine max*	Antiglycation, hypoglycemic, inhibition of HbA1c, Anti-atherosclerotic activity, pancreatic β cell regeneration, antiglycation, antioxidant system activation	Impaired insulin production, Abnormal urine glucose, polyuria, polydipsia	GLUT4 activation, glucose-metabolizing enzyme upregulation	[131,132]
Carnosine	β-alanyl-L-histidine	Vertebrate tissues	Bivalent metal ion chelation, proton buffering, and carbonyl scavenger activities	Diabetes, aging	Antioxidant pathway activation	[133,134]
Polypeptide-p	Peptide	*Momordica charantia*	Hypoglycemic	Diabetes and associated complications	Glucose-metabolizing enzyme upregulation	[135,136]

## Data Availability

Not applicable.

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
