# Peer review of "Anti-Glucotoxicity Effect of Phytoconstituents via Inhibiting MGO-AGEs Formation and Breaking MGO-AGEs"

_ijms, 2023, doi:10.3390/ijms24087672_

Round 1

Reviewer 1 Report

This review is through and comprehensive but it is also repetitive in places, could be reduced in length and could also be improved by the adjustment of some sections of text (as indicated below)

Main suggestions:

Introduction, end of paragraph 1. 'In recent years, certain newly discovered....from plants demonstrated more effectiveness in treating diabetes than oral hypoglycemic medications...'.   This is a bold and strong statement that is not supported by any references. I feel it is too bold a statement at this time in the field and doesn't align well with the next sentence indicating traditional medicine has a promising future.  I feel the sentence should be softened in its statement regarding plant compounds being more effective than oral medications - suggest the emphasis should be on 'promising data/evidence being revealed which is encouraging'

Section 3 could be improved by reducing its size and focusing. Section 3 (and some other sections) have discussion of the effects of some phytochemical/natural compounds on AGEs etc - these sections are better moved to the dedicated Section 5

Section 5.5.4 is quite repetitive of section 5.4.1. Suggest move section 5.5.4 to 5.4.1

Both tables could be in a smaller font to reduce the text changes and make it easier to read/digest

There is a section on flavonoids (5.4) and yet an abundant class of flavonoids anthocyanins is not reviewed. Add a section specifically on anthocyanins and their glycosides and the evidence and promise for managing MGO and AGEs

Figure 1.  Figure 1 is referenced in the section on resveratrol yet the figure is quite generic in indicating the mechanism of action of AGEs from a broad range of phytoconstituents - hence I feel Fig 1 should be referenced and described toward the end of the review, maybe in the conclusions Section 6.

Minor errors spotted:

Introduction, line 4 - 'roasted'

Section 2, line 9 - 'These ca be identified'.... can

Section 5.5.4.  Spelling of proanthocyanidins and procyanidins needs fixing in this section

Author Response

Response to Reviewer 1 comments

Comment 1: Introduction, end of paragraph 1. 'In recent years, certain newly discovered....from plants demonstrated more effectiveness in treating diabetes than oral hypoglycemic medications...'.   This is a bold and strong statement that is not supported by any references. I feel it is too bold a statement at this time in the field and doesn't align well with the next sentence indicating traditional medicine has a promising future.  I feel the sentence should be softened in its statement regarding plant compounds being more effective than oral medications - suggest the emphasis should be on 'promising data/evidence being revealed which is encouraging'

Response: Thank you for your kind suggestion. The sentence has been modified as per your suggestion.

Comment 2: Section 3 could be improved by reducing its size and focusing. Section 3 (and some other sections) have discussion of the effects of some phytochemical/natural compounds on AGEs etc - these sections are better moved to the dedicated Section 5

Response: Thank you for your valuable suggestion. However, it’s not possible to move the parts of section 3 to section 5 as section 5 and its subsections discuss about specific phytochemicals and their effect in disease prevention and treatment separately.

Comment 3: Section 5.5.4 is quite repetitive of section 5.4.1. Suggest move section 5.5.4 to 5.4.1

Response: Thank you for your kind suggestion. However, sections 5.4.1 and 5.5.4 describe different phytochemicals with similar name. Section 5.4.1 describes about Procyanidins whereas, section 5.5.4.  describes about Proanthocynidins and Pocynidin B2. Also, chemically, these are different that’s why these are part of separate sections.

Comment 4: Both tables could be in a smaller font to reduce the text changes and make it easier to read/digest

Response: Thank you for your kind suggestion. Font in the table are as per the journal format.

Comment 5: There is a section on flavonoids (5.4) and yet an abundant class of flavonoids anthocyanins is not reviewed. Add a section specifically on anthocyanins and their glycosides and the evidence and promise for managing MGO and AGEs

Response: Thank you for your valuable suggestion. However, we have planned our future research specifically on the role of anthocyanins and their glycosides in MGO and AGEs mediated disease conditions. Because this class of phytoconstituents is very diverse and contains lot of information. Adding anthocyanins in this review will more extend the length of the paper unnecessarily.

Comment 6: Figure 1.  Figure 1 is referenced in the section on resveratrol yet the figure is quite generic in indicating the mechanism of action of AGEs from a broad range of phytoconstituents - hence I feel Fig 1 should be referenced and described toward the end of the review, maybe in the conclusions Section 6.

Response: Thank you for your kind suggestion. In section 5, we have emphasized on various phytoconstituents and their role in AGEs inhibition. We, thought referencing fig 1 in section 5 is suitable and usually conclusion section avoids figure referencing.

Minor errors spotted:

Comment 7: Introduction, line 4 - 'roasted'

Response: The text has been corrected.

Comment 8: Section 2, line 9 - 'These ca be identified'.... can

Response: The text has been corrected.

Comment 9: Section 5.5.4.  Spelling of proanthocyanidins and procyanidins needs fixing in this section

Response: The text has been corrected.

Reviewer 2 Report

1. The contents in the Tables need to be rearranged so that readers can easily understand. The overall layout and content are somewhat chaotic, especially in Table 2.

2. Figure 1 is too simple. There are already many AGE-related signaling pathways and types of molecules/factors. Therefore, it should be strengthened in Figure 1.

3. The paragraph lengths in the text are too long. The review article should have effective classification and paragraphs. Subheadings should be used to facilitate the understanding of the classification and key points by readers.

Author Response

Response to Reviewer 2 comments

Comment 1. The contents in the Tables need to be rearranged so that readers can easily understand. The overall layout and content are somewhat chaotic, especially in Table 2.

Response: Thank you for your valuable suggestion. The contents in table 2 have rearranged. 

Comment 2. Figure 1 is too simple. There are already many AGE-related signaling pathways and types of molecules/factors. Therefore, it should be strengthened in Figure 1.

Response: Thank you for your kind suggestion. Different molecules/factors usually activate some common signaling pathways and adding all of them at one place will make hard to understand the exact mechanism of inhibition of AGEs induced disease conditions by phytoconstituents. So, we intentionally aimed to keep fig1 simple for better understanding of the reader.

Comment 3. The paragraph lengths in the text are too long. The review article should have effective classification and paragraphs. Subheadings should be used to facilitate the understanding of the classification and key points by readers.

Response: Thank you for suggestion. The text in paragraphs have been edited and subheadings have been used for better understanding of the classification and key points by readers.

Reviewer 3 Report

Phytochemicals have therapeutic benefits for various illnesses and show promise in creating new medications. Traditional medicines based on phytoconstituents have anti-biotic, anti-oxidant, and wound-healing effects. These elements are crucial for fighting pathological conditions and accelerating wound healing. The review of 221 research papers discusses the role of phytoconstituents in breaking down AGEs and scavenging MGO, highlighting their potential in developing functional foods for health benefits.

The premise of the argument is intriguing; however, the writing requires some adjustments. Several phrases and concepts are repeated in slightly different forms, which impedes readability and unnecessarily lengthens the text. Additionally, certain ideas are presented in a convoluted wordy manner, which further hinders the overall flow and coherence of the manuscript. In light of these concerns, a comprehensive restructuring of the manuscript is recommended to ensure a more organized and lucid presentation of the ideas being conveyed.
Moreover, I suggest adding a cartoon explaining the fundamental AGEs/RAGE pathway and its interleaving with the environment. (see some questions for clarification listed below)

One of the main characteristics of AGEs (and ALEs) compounds is that the process is random: On the basis of their chemical environment, statistically, some lysine or arginine, are located in a more favorable position to the glycation than others; but this does not ensure that glycation happened nor if it was, that the glycation ever happens in the same way, indeed, the process of glycation is at random. Please highlight this concept.

The cascade of pathological events triggered by advanced glycation end products could play a

crucial role in the hyperglycemia‐independent complications of diabetes.
There are only a few data available in the literature studying the effects induced by glyco‐oxidation

either on preadipocytes or on adipocytes and describing their impact on differentiation and

proliferation. cit: Chen CY, Abell AM, Moon YS, Kim KH. An advanced glycation end product (AGE)‐receptor for AGEs (RAGE) axis restores the adipogenic potential of senescent preadipocytes through modulation of p53 protein function. The Journal of biological chemistry. 2012;287(53):44498‐507.

In the vast majority of cases, the glycating insult is supplied only for a short time window (e.g. 30min ‐ 24/48h) which hardly can mimic the long‐lasting in vivo exposure of adipose tissue cellular components to AGEs.

Instead, this milestone groundbreaking study which analyzes the exposure to the glycation‐inducing environment during all the adipogenic process reflects the real situation in the body
In vitro chronic glycation induces AGEs accumulation reducing insulin stimulated glucose uptake and increasing GLP1R in adipocytes; Am J Physiol Endocrinol Metab. 2021 Mar 29. doi: 10.1152/ajpendo.00156.2020.”

Line 214-215: Recent research has shown that the gut microbiota regulates an aging-related loss in gut barrier integrity, allowing AGEs to escape from the gut into the bloodstream and compromising brain microglial activity.
what do you mean by “escape”? What is the role of gut microbiota?

Apart from the fluorescent property of the AGEs in question, what other significant differences exist, and how do these differences affect health? Entering in a cell a fluorescent compared to a nonfluorescent one, what consequences will it bring?

Line 281: “The genetic capacity to detoxify mechanisms against the accumulation of AGEs may also have an impact on variations in circulating AGEs” what do you mean by <genetic capacity>?

Line 324 – 341, I think some references are missing, please recheck.
line 341: Hb-AGEs also induced ED 341 by inhibiting migration and proliferation of HUVECs. Do you mean Endothelial cell dysfunction with ED? Please specify.
Line 408: “Cellular movement and the increase of proinflammatory and prothrombotic molecules are mediated by AGEs-RAGE interaction” in what way?

Plant extracts have shown remarkable efficacy in inhibiting and preventing glycation. However, is the precise mechanism of action underlying their therapeutic effects well understood? Could you explain the general mechanism?

line 575: “It stimulates GLUT4 translocation to the cell membrane……..” this sentence could be discussed in the light of the influence of AGEs on GLUT4 translocation to the plasma membrane and/or on its functionality.

Line 594-595: “High glucose levels are necessary for these antihyperglycemic, insulinotropic, and glucagonostatic actions, which are essentially plasma glucose level dependent”
could you rephrase?

Line 644: “has revealed that CA can inhibit AGEs induced fluorescence in vitro…” again, what is the meaning of the fluorescence? It is indicator, of what?

Line 927: Manuka Honey is known for its potent antibacterial properties, which are attributed to its high concentration of methylglyoxal (MGO)
line 934 - “The antioxidant properties of honey reduce the formation of advanced glycation and lipoxidation end products, which can induce inflammation and 935 oxidative stress in diabetics [208]. The antioxidant, anti-inflammatory, enhanced wound healing and antibacterial activity of MH is due to MGO content in it”
could you explain to me the effect of MGO and that specific honey?

The conclusions drawn from the study are characterized by disorganization and lack of clarity. Specifically, it is unclear how the natural products in question exert their effects, including the timing and dosage required for optimal efficacy. To address these gaps in knowledge, further research is necessary. Do you have any suggestions on what to investigate?

minor:
acronymous and references are not uniform throughout the manuscript.

Author Response

Response to Reviewer 3 comments

Comment 1: Phytochemicals have therapeutic benefits for various illnesses and show promise in creating new medications. Traditional medicines based on phytoconstituents have anti-biotic, anti-oxidant, and wound-healing effects. These elements are crucial for fighting pathological conditions and accelerating wound healing. The review of 221 research papers discusses the role of phytoconstituents in breaking down AGEs and scavenging MGO, highlighting their potential in developing functional foods for health benefits. The premise of the argument is intriguing; however, the writing requires some adjustments. Several phrases and concepts are repeated in slightly different forms, which impedes readability and unnecessarily lengthens the text. Additionally, certain ideas are presented in a convoluted wordy manner, which further hinders the overall flow and coherence of the manuscript. In light of these concerns, a comprehensive restructuring of the manuscript is recommended to ensure a more organized and lucid presentation of the ideas being conveyed.

Response: Thank you for your valuable suggestions. The desired changes have been made as per the suggestions of the eminent reviewer.

Comment 2: Moreover, I suggest adding a cartoon explaining the fundamental AGEs/RAGE pathway and its interleaving with the environment.

Response: Thank you for your kind suggestion. However, I have just relocated to another place and I am not in the condition to draw a cartoon as suggested by the eminent reviewer due to shifting.

Comment 3: One of the main characteristics of AGEs (and ALEs) compounds is that the process is random: On the basis of their chemical environment, statistically, some lysine or arginine, are located in a more favorable position to the glycation than others; but this does not ensure that glycation happened nor if it was, that the glycation ever happens in the same way, indeed, the process of glycation is at random. Please highlight this concept.

Response: Thank you for your valuable suggestions. We have highlighted the text as per your suggestion. It has been added to the second paragraph of the introduction section (line 73-81).

“The process of glycation is typically random. Glycation can change the primary, secondary and tertiary structures of the proteins that can create lysine-arginine cross-links, which cause protein aggregation. Protein aggregation can affect the feasibility of lysine or arginine groups in the protein that can change the overall glycation process. Additionally, the binding of certain groups at the charged side chain of lysine can also affect the rate of glycation. For example, the attachment of carbohydrate on the lysine side chain by glycation might have an even larger effect because not only the charge, but also the size of the side chain of lysine is modified”.

Comment 4: The cascade of pathological events triggered by advanced glycation end products could play a crucial role in the hyperglycemia‐independent complications of diabetes.
There are only a few data available in the literature studying the effects induced by glyco‐oxidation

either on preadipocytes or on adipocytes and describing their impact on differentiation and

proliferation. cit: Chen CY, Abell AM, Moon YS, Kim KH. An advanced glycation end product (AGE)receptor for AGEs (RAGE) axis restores the adipogenic potential of senescent preadipocytes through modulation of p53 protein function. The Journal of biological chemistry. 2012;287(53):44498507.

In the vast majority of cases, the glycating insult is supplied only for a short time window (e.g. 30min ‐ 24/48h) which hardly can mimic the long‐lasting in vivo exposure of adipose tissue cellular components to AGEs.

Instead, this milestone groundbreaking study which analyzes the exposure to the glycation‐inducing environment during all the adipogenic process reflects the real situation in the body
In vitro chronic glycation induces AGEs accumulation reducing insulin stimulated glucose uptake and increasing GLP1R in adipocytes; Am J Physiol Endocrinol Metab. 2021 Mar 29. doi: 10.1152/ajpendo.00156.2020.”

Line 214-215: Recent research has shown that the gut microbiota regulates an aging-related loss in gut barrier integrity, allowing AGEs to escape from the gut into the bloodstream and compromising brain microglial activity.
what do you mean by “escape”? What is the role of gut microbiota?

Response: Thank you for your suggestion. The text has been edited for better understanding and also the role of gut microbiota has been clarified. “Aging usually brings rapid changes and deterioration in the gut microbiome that results in an increased intestinal permeability and thereby, increased absorption of AGEs into the bloodstream that allows accumulation of AGEs in microglial tissue resulting in declined microglial function” (line 226-229).

Comment 7: Apart from the fluorescent property of the AGEs in question, what other significant differences exist, and how do these differences affect health? Entering in a cell a fluorescent compared to a nonfluorescent one, what consequences will it bring?

Response: Apart from the fluorescent property, AGEs possess different cross-linking properties based on their chemical structure. Non-fluorescent no-crosslinking AGEs are formed much faster than other AGEs therefore, their concentration is usually higher. Moreover, fluorescent properties of AGEs can be utilized to measure AGE (biomarker) as skin autofluorescence (SAF) by the AGE Reader.

Comment 8: Line 281: “The genetic capacity to detoxify mechanisms against the accumulation of AGEs may also have an impact on variations in circulating AGEs” what do you mean by <genetic capacity>?

Response: Changes in gene expression of transcription factors related to detoxify mechanisms against AGEs accumulation in cellular environment.

Comment 9: Line 324 – 341, I think some references are missing, please recheck.

Response: The text from line 324-341 has been checked and the relevant reference has been mentioned at the end of the description.

Comment 10: line 341: Hb-AGEs also induced ED 341 by inhibiting migration and proliferation of HUVECs. Do you mean Endothelial cell dysfunction with ED? Please specify.

Response: ED means endothelial dysfunction.

Comment 11: Line 408: “Cellular movement and the increase of proinflammatory and prothrombotic molecules are mediated by AGEs-RAGE interaction” in what way?

Response: RAGE is found on a variety of cell types involved in the immune inflammatory response. The interaction between AGEs and RAGE on cells, including monocytes, macrophages, and endothelial cells, can control cellular migration and the upregulation of proinflammatory and prothrombotic molecules.

Comment 12: Plant extracts have shown remarkable efficacy in inhibiting and preventing glycation. However, is the precise mechanism of action underlying their therapeutic effects well understood? Could you explain the general mechanism?

Response: The general mechanisms of inhibiting and preventing glycation by plant extracts include inhibition of ROS, dicarbonyl trapping, activation of antioxidants, and disruption of protein cross-linking.

Comment 13: line 575: “It stimulates GLUT4 translocation to the cell membrane……..” this sentence could be discussed in the light of the influence of AGEs on GLUT4 translocation to the plasma membrane and/or on its functionality.

Response: The discussion has been improved highlighting the effect of AGEs on GLUT4 translocation to the plasma membrane (line 587-589).

Comment 14: Line 594-595: “High glucose levels are necessary for these antihyperglycemic, insulinotropic, and glucagonostatic actions, which are essentially plasma glucose level dependent”
could you rephrase?

Response: The sentence has been rephrased as per your kind suggestion (line 609-610).

Comment 15: Line 644: “has revealed that CA can inhibit AGEs induced fluorescence in vitro…” again, what is the meaning of the fluorescence? It is indicator, of what?

 Response: Yes, fluorescence property of AGEs serves as indicator of their presence in cells. CA can inhibit AGEs formation and thereby, AGEs induced fluorescence.

Comment 16: Line 927: Manuka Honey is known for its potent antibacterial properties, which are attributed to its high concentration of methylglyoxal (MGO).

line 934 - “The antioxidant properties of honey reduce the formation of advanced glycation and lipoxidation end products, which can induce inflammation and 935 oxidative stress in diabetics [208]. The antioxidant, anti-inflammatory, enhanced wound healing and antibacterial activity of MH is due to MGO content in it” could you explain to me the effect of MGO and that specific honey?

 Response: MGO in manuka honey is its natural component and it has several therapeutic properties including tissue regeneration and wound healing. Moreover, it can prevent the biofilm formation (slimy layer) that bacteria build around themselves to protect them from antibiotics.

Comment 18: The conclusions drawn from the study are characterized by disorganization and lack of clarity. Specifically, it is unclear how the natural products in question exert their effects, including the timing and dosage required for optimal efficacy. To address these gaps in knowledge, further research is necessary. Do you have any suggestions on what to investigate?

 Response: Thank you for your valuable suggestion. Further research on identifying the most effective phytoconstituents and using them to prepare functional drugs or foods that can reduce AGEs mediated harmful effects is required. Clinical trials to examine the timing and dosage as well as pharmacological effects and well-designed trials in humans are required.

minor:
Comment 19: acronymous and references are not uniform throughout the manuscript.

Response: Acronymous and references have been checked and edited as per your kind suggestion.

Round 2

Reviewer 2 Report

I did not observe any modifications made to the manuscript by the authors in accordance with the comments. On the contrary, the manuscript is almost identical to the initial submission. It is recommended that the authors revise the manuscript as per the comments provided. Additionally, Table 1 has not been adequately improved and has become scrambled data.

Author Response

Response to Reviewer 2 comments (R2)

Comment 1: I did not observe any modifications made to the manuscript by the authors in accordance with the comments. On the contrary, the manuscript is almost identical to the initial submission. It is recommended that the authors revise the manuscript as per the comments provided.

Response: Thank you for your valuable suggestion and recommendations. The manuscript has been revised as per the suggestion of the eminent reviewer. The text in the paragraphs have been modified and the long sentences in different sections have been edited for better understanding by the readers.

Comment 2: Additionally, Table 1 has not been adequately improved and has become scrambled data.

Response: Table 1 has been edited and improved.

Response to Reviewer 2 comments (R1)

Comment 1. The contents in the Tables need to be rearranged so that readers can easily understand. The overall layout and content are somewhat chaotic, especially in Table 2.

Response: Thank you for your valuable suggestion. The contents in table 1 and table 2 have been rearranged. 

Comment 2. Figure 1 is too simple. There are already many AGE-related signaling pathways and types of molecules/factors. Therefore, it should be strengthened in Figure 1.

Response: Thank you for your kind suggestion. The figure 1 has been strengthened as per the suggestion of eminent reviewer.  

Comment 3. The paragraph lengths in the text are too long. The review article should have effective classification and paragraphs. Subheadings should be used to facilitate the understanding of the classification and key points by readers.

Response: Thank you for suggestion. The text in paragraphs have been edited and subheadings have been used for better understanding of the classification and key points by readers.

Reviewer 3 Report

authors have responded to all my comments, I have only a few small notes to be added.

-      Line 306: please use the description you gave me in your response to the reviews of “genetic capacity”.

-      Line 609-610

<AGEs have been found to inhibit Glut-4 translocation from the cytoplasm to the plasma membrane>
this reference, about the pioneering  in vitro study that elucidate that mechanism, is missing: “
In vitro chronic glycation induces AGEs accumulation reducing insulin stimulated glucose uptake and increasing GLP1R in adipocytes.” Chilelli, N. C., Faggian, A. et al. 2021

-      Line 601-607

<These are a type of triterpenoid found in Momordica charantia (bitter melon) and

are of many types such as (A, B, C, D, E, G, F1, F2, I, K, L) Momordicosides (Q, R, S,

U, and T) have been found advantageous for diabetic people.>
 how many Momordicosides in total are known? Which is their common structure?

>>> 

Comment 7: Apart from the fluorescent property of the AGEs in question, what other significant differences exist, and how do these differences affect health? Entering in a cell a fluorescent compared to a nonfluorescent one, what consequences will it bring?

Response: Apart from the fluorescent property, AGEs possess different cross-linking properties based on their chemical structure. Non-fluorescent no-crosslinking AGEs are formed much faster than other AGEs therefore, their concentration is usually higher. Moreover, fluorescent properties of AGEs can be utilized to measure AGE (biomarker) as skin autofluorescence (SAF) by the AGE Reader.

>>>>>>>

I found this answer really interesting, thank you, and please add it to the main text.

Author Response

Response to Reviewer 3 comments

Comment 1: authors have responded to all my comments, I have only a few small notes to be added.

Line 306: please use the description you gave me in your response to the reviews of “genetic capacity”.

Response: Thank you for your valuable suggestion. The text has been edited as per the suggestion of the eminent reviewer (line 318-320).

Comment 2: - Line 609-610

<AGEs have been found to inhibit Glut-4 translocation from the cytoplasm to the plasma membrane>this reference, about the pioneering in vitro study that elucidate that mechanism, is missing: “In vitro chronic glycation induces AGEs accumulation reducing insulin stimulated glucose uptake and increasing GLP1R in adipocytes.” Chilelli, N. C., Faggian, A. etal. 2021

Response: The reference has been added (line 618).

Comment 3: - Line 601-607

<These are a type of triterpenoid found in Momordica charantia(bitter melon) and

are of many types such as (A, B, C, D, E, G, F1, F2, I, K, L) Momordicosides (Q, R, S,

U, and T) have been found advantageous for diabetic people.> how many Momordicosides in total are known? Which is their common structure?

Response: about 200 reported structures so far.

The common structure of momordicosides is 2-methyl-6-(4,4,9,13,14-pentamethyl-3-((3,4,6-trihydroxy-5-((3,4,5-trihydroxytetrahydro-2H-pyran-2-yl)oxy)tetrahydro-2H-pyran-2-yl)oxy)-2,3,4,7,8,9,10,11,12,13,14,15,16,17-tetradecahydro-1H-cyclopenta[a]phenanthren-17-yl)heptane-2,3,4,5-tetraol.

>>>

Comment 4: Apart from the fluorescent property of the AGEs in question, what other significant differences exist, and how do these differences affect health? Entering in a cell a fluorescent compared to a nonfluorescent one, what consequences will it bring?

Response: Apart from the fluorescent property, AGEs possess different cross-linking properties based on their chemical structure. Non-fluorescent no-crosslinking AGEs are formed much faster than other AGEs therefore, their concentration is usually higher. Moreover, fluorescent properties of AGEs can be utilized to measure AGE (biomarker) as skin autofluorescence (SAF) by the AGE Reader.

>>>>>>>I found this answer really interesting, thank you, and please add it to the main text.

Response: Thank you for your kind suggestion. The text has been added to the main text (line 253-258).

Round 3

Reviewer 2 Report

No further suggestion or comment